# Nitroglycerin-responsive gene switch for the on-demand production of therapeutic proteins

Mohamed Mahameed[1,2,6], Shuai Xue [1,3,6], Benjamin Danuser[1], Ghislaine Charpin-El Hamri[4], Mingqi Xie [3] & Martin Fussenegger [1,5] ✉

Gene therapies and cell therapies require precise, reversible and patient-friendly control over the production of therapeutic proteins. Here we present a fully human nitric-oxide-responsive gene-regulation system for the on-demand and localized release of therapeutic proteins through clinically licensed nitroglycerin patches. Designed for simplicity and robust human compatibility, the system incorporates human mitochondrial aldehyde dehydrogenase for converting nitroglycerin into nitric oxide, which then activates soluble guanylate cyclase to produce cyclic guanosine monophosphate, followed by protein kinase G to amplify the signal and to trigger target gene expression. In a proof-of-concept study, human cells expressing the nitroglycerin-responsive system were encapsulated and implanted subcutaneously in obese mice with type 2 diabetes. Transdermal nitroglycerin patches applied over the implant enabled the controlled and reversible production of glucagon-like peptide-1 throughout the 35-day experimental period, effectively restoring blood glucose levels in these mice without affecting heart rate or blood pressure. The approach may facilitate the development of safe, convenient and responsive implantable devices for the sustained delivery of biopharmaceuticals for the management of chronic diseases.

With gene therapies and cell therapies gathering momentum as the next pillar in modern personalized medicine, the quest is ongoing for efficient gene switches to provide trigger-inducible dynamic in situ production and systemic delivery of biopharmaceuticals in a robust, reliable and tunable manner while maximizing patient convenience and compliance. For this purpose, we require bioavailable and easy-to-administer trigger compounds devoid of pleiotropic effects together with compact, orthogonal and fully human control components that can provide long-term, reversible, adjustable and dynamic control of transgene expression with minimum side effects or interference with host

metabolism. Meeting all these characteristics to a high standard remains an ongoing challenge. Various trigger compounds, including clinically licensed drugs[1,2], food additives[3], cosmetics[4] and fragrances[5], have been successfully used to stimulate protein pharmaceutical release. In addition, non-molecular traceless physical cues such as light[6], electricity[7], radio waves[8], heat[9] or sound[10] have been considered as control inputs to alleviate complications associated with insufficient bioavailability, pleiotropic effects and poor pharmacodynamics. The utility of gene switches employing many of these inducers has been demonstrated in the experimental control of diverse medical conditions, including cancer[11],

[1]Department of Biosystems Science and Engineering, ETH Zurich, Basel, Switzerland. [2]Institute for Drug Research (IDR), School of Pharmacy, Faculty of Medicine, The Hebrew University of Jerusalem, Jerusalem, Israel. [3]Westlake Laboratory of Life Sciences and Biomedicine, Hangzhou, China. [4]Département Génie Biologique, Institut Universitaire de Technologie, Université Claude Bernard Lyon 1, Villeurbanne Cedex, France. [5]Faculty of Science, University of Basel, Basel, Switzerland. [6]These authors contributed equally: Mohamed Mahameed, Shuai Xue. ✉e-mail: fussenegger@bsse.ethz.ch

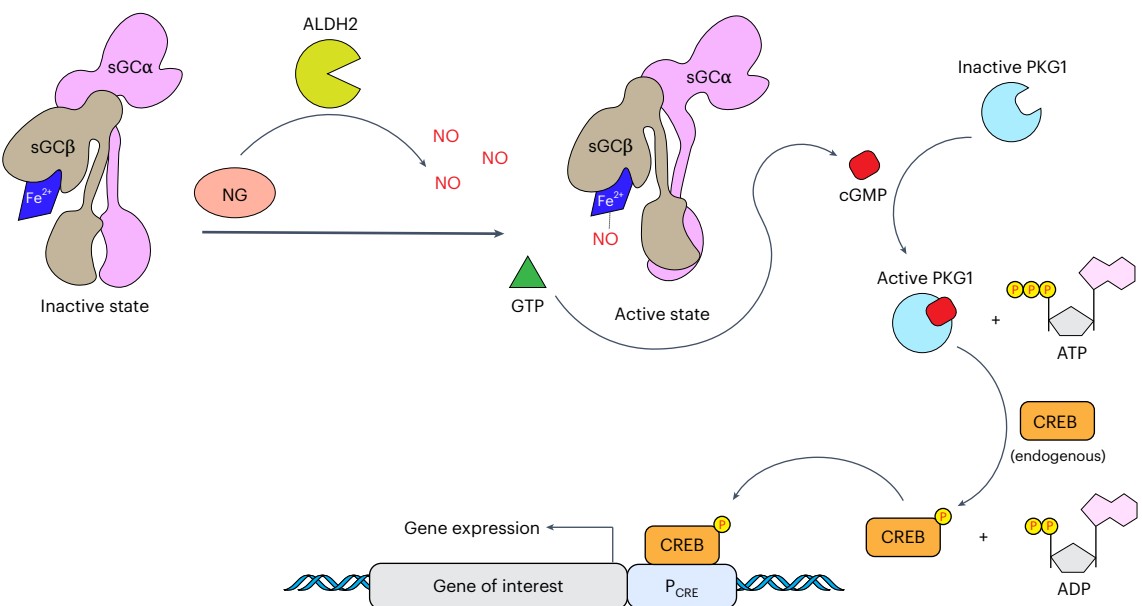

**Fig. 1 | Schematic illustration showing the molecular components and mechanism of action of hNORM in mammalian cells.** Administered NG is enzymatically metabolized inside the cells by ALDH2, leading to the generation of NO. Produced NO molecules activate the sGC αβ heterodimer by binding to the ferrous ion ($Fe^{2+}$) of the prosthetic haem group. Enzymatically active sGC catalyses the production of the secondary messenger cGMP from the precursor GTP. High levels of cGMP bind to and activate cGMP-dependent PKG1, which subsequently phosphorylates the endogenous transcription factor CREB. PKG1 utilizes adenosine triphosphate (ATP) as a phosphate and energy source, generating adenosine diphosphate (ADP) as a by-product. Following its phosphorylation, CREB binds to a synthetic promoter ($P_{CRE}$) that contains CRE and initiates the expression of the gene of interest.

metabolic disorders[7] and bacterial infections[12]. Nevertheless, there is still a need for fully humanized gene switches driven by safe, convenient, clinically licensed small-molecular compounds that meet all of the above criteria.

In this context, we focused on nitroglycerin (NG), which has been widely used for over 130 years as a vasodilator to treat heart conditions such as angina pectoris and chronic heart failure[13,14]. NG is commercially available as intravenous solutions, sublingual tablets, sprays, ointments and transdermal patches[15]. The therapeutic impact of NG unfolds after its conversion to nitric oxide (NO), which is mainly mediated by mitochondrial aldehyde dehydrogenase 2 (ALDH2)[16,17]. NO is a gaseous signalling molecule involved in controlling a variety of central biological processes, including cardiovascular activities, immunity and central nervous system homeostasis[18]. With a lifetime of just a few seconds and being freely diffusible across biological membranes, NO appears to be a promising local, transient, autocrine and paracrine biological messenger, which is rapidly inactivated by conversion to nitrates and nitrites in the presence of oxygen and water[19,20]. In mammalian cells, NO is biosynthesized from L-arginine, oxygen and NADPH by various NO synthases[21]. In humans, the endothelium of blood vessels releases NO to trigger relaxation of the surrounding muscle cells, resulting in vasodilation and increased blood flow[13]. In the central nervous system, NO regulates synaptic plasticity, the sleep–wake cycle and hormone secretion[22], while in the immune system, NO is involved in controlling immunity and inflammation and serves as a defence metabolite against invading pathogens[23]. The primary target of NO in mammalian cells is soluble guanylate cyclase (sGC), a heterodimer composed of alpha and beta subunits, each of which consists of an N-terminal haem nitric oxide/oxygen (HNOX) binding domain, a PAS-like domain that facilitates heterodimer formation via the coiled-coil domains and propagates a signal to the C-terminal catalytic domain, which produces cyclic guanosine monophosphate (cGMP) from guanosine triphosphate (GTP) after binding of NO to the HNOX haem[24]. As a key second messenger, cGMP triggers specific signalling cascades that regulate the function and expression of diverse cell-type-dependent sets of proteins and genes[25].

Owing to the critical functions of NO in central activities of the human body, NO-controlled signalling pathways have been successfully targeted by various drugs primarily designed for the treatment of hypertension, chronic heart failure and erectile dysfunction[18]. Apart from direct inhalation of NO in newborns suffering from persistent pulmonary hypertension[26], direct administration of NO gas remains challenging, in particular owing to its gaseous state, short half-life[19] and difficulty of precise dosing, which increases the risk of life-threatening toxicity[27,28]. To overcome these issues, NO-donor prodrugs, such as the vasodilators NG, sodium nitroprusside and molsidomine, which provide sustained release of NO during their metabolism in the body, have been developed and successfully used for decades[29,30]. Among these drugs, only NG shows almost complete bioavailability when transdermally applied[31]. Owing to NG's short half-life of approximately 5 min in the body, NG-delivering transdermal patches have become the standard for the management of angina pectoris[32].

The aim of this work was to capitalize on these properties of NO and NG by designing a fully human nitric oxide-responsive transgene regulation modality (hNORM) to enable dynamic control of in situ biopharmaceutical production and real-time systemic dosing by means of patch-based percutaneous delivery of NG, thereby propelling the use of NG, which has been validated for safe use in humans for over 130 years, into the gene- and cell-based therapy era. As illustrated schematically in Fig. 1, the developed hNORM system utilizes human mitochondrial ALDH2 to ensure efficient local conversion of percutaneously delivered NG into the trigger compound NO. In human endothelial and smooth muscle cells, native NO signalling is triggered by the binding of NO to sGC, which catalyses the conversion of GTP into the intracellular signalling molecule cGMP. Therefore, to equip other human cell types with NO-signalling ability and sensitize them to NO-triggered control of transgene expression, we introduced human heterodimeric ferrous haem-containing sGC for NO-dependent production of the second messenger cGMP. Specifically, we ectopically expressed the alpha and beta subunits of sGC (pMMH178, $P_{hPGK}$-sGCα-pA_{bGH}; pMMH179, $P_{hPGK}$-sGCβ-pA_{bGH}). We also introduced human cGMP-dependent protein

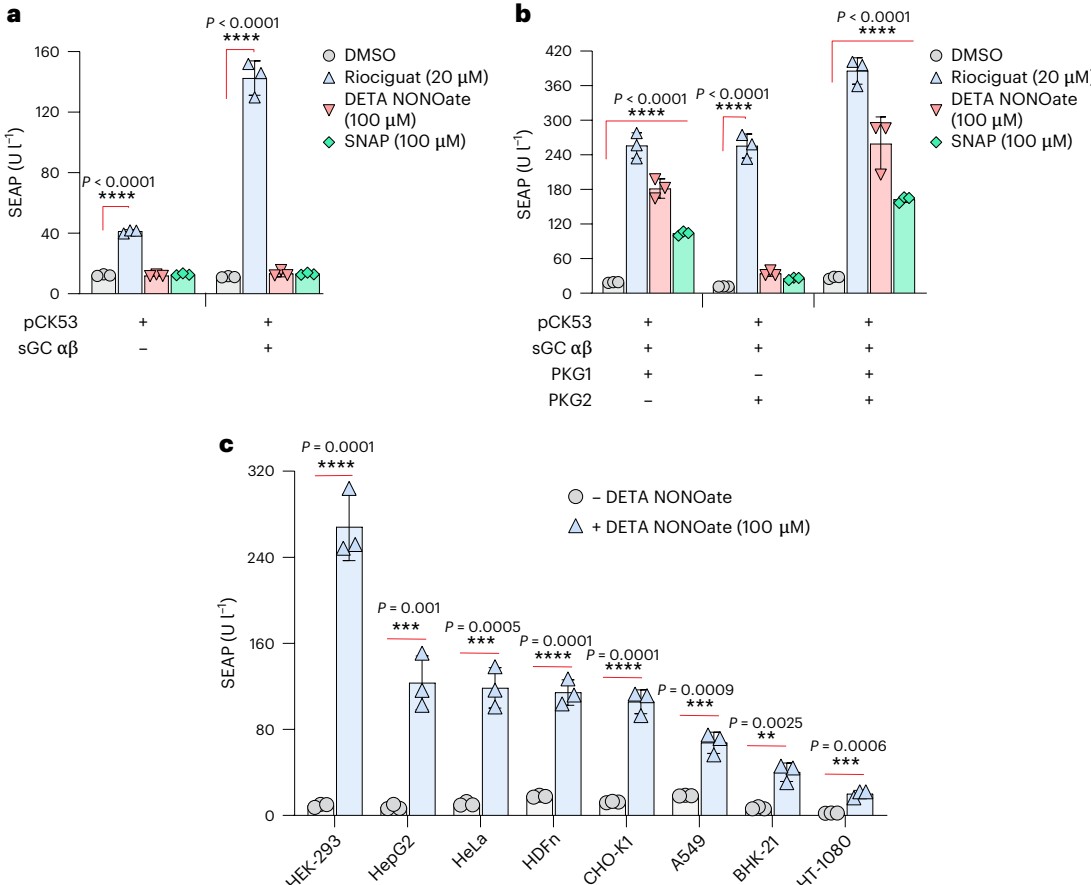

**Fig. 2 | Engineering of hNORM in mammalian cells. a**, SEAP levels in culture supernatants of HEK-293 cells transfected with pMMH178 ($P_{hPGK}$-sGCα-pA$_{bGH}$) and pMMH179 ($P_{hPGK}$-sGCβ-pA$_{bGH}$) along with the reporter plasmid pCK53 ($P_{CRE}$-SEAP-pA$_{bGH}$). At 24 h following the transfection, the medium was changed to 100 μl of fresh medium containing either DMSO, riociguat (20 μM), DETA NONOate (100 μM) or SNAP (100 μM) for 24 h. Data are presented as mean ± s.d. ($n = 3$), and $P$ values were calculated using a two-tailed, unpaired Student's $t$-test. **b**, SEAP levels in culture supernatants of HEK-293 cells transfected with the plasmids described in the previous experiment in addition to the co-transfection of PKG1 WT ($P_{hCMV}$-PKG1β-pA$_{bGH}$) and/or PKG2 ($P_{hCMV}$-PKG2-pA$_{bGH}$). Data are presented as mean ± s.d. ($n = 3$), and $P$ values were calculated using a two-tailed, unpaired Student's $t$-test. **c**, The performance of hNORM in various mammalian cells. HEK-293, HepG2, HeLa, HDFn, CHO-K1, A549, BHK-21 and HT-1080 cells were co-transfected with pMMH178, pMMH179, PKG1β WT and pCK53. Following the transfection, DETA NONOate (100 μM) was used as an inducer for 24 h as described for the previous experiments. Data are presented as mean ± s.d. ($n = 4$), and $P$ values were calculated using a two-tailed, unpaired Student's $t$-test. All data shown are biological replicates. Source data are provided as a Source Data file.

kinase G1 (PKG1), which amplifies the cGMP signal and phosphorylates cAMP-responsive element binding protein (CREB)[33]. Phosphorylated CREB in turn transactivates synthetic promoters ($P_{CRE}$) containing cAMP-response elements (CRE), thereby rewiring the NG-triggered NO sensing to transgene expression.

As a proof of concept, we show that percutaneous delivery of NG from clinically licensed dermal patches placed above subcutaneously implanted microencapsulated hNORM-transgenic human cells provides precise, reversible and dynamic NO-triggered control of the expression of the therapeutic protein GLP-1, and can successfully treat experimental type 2 diabetes and obesity in mice over a prolonged period, without affecting parameters such as heart rate and blood pressure. We believe that the introduction of patch-based transdermal control over cell-based therapies will increase acceptance, convenience and compliance in personalized health interventions for the treatment of chronic medical conditions.

## Results

### hNORM in mammalian cells

We began the development of the hNORM system by examining the effect of NO in HEK-293 cells co-transfected with pMMH178 ($P_{hPGK}$-sGCα-pA$_{bGH}$) and pMMH179 ($P_{hPGK}$-sGCβ-pA$_{bGH}$) to express

the intact sGC heterodimer. To monitor performance, we rewired cGMP signalling via CREB to a synthetic promoter containing cAMP/cGMP-response elements ($P_{CRE}$) (pCK53, $P_{CRE}$-SEAP-pA$_{bGH}$) to drive the expression of human placental secreted alkaline phosphatase (SEAP), the activity of which can be easily measured. For convenience in the early design stages, we used the NO donors DETA NONOate[34] and SNAP ($S$-nitroso-$N$-acetylpenicillamine)[35], which have half-lives of several hours, to generate NO in the cell cultures in a precise and reproducible manner. The non-NO donor and clinically licensed anti-hypertension drug riociguat was also included as an sGC positive-control inducer[36]. However, in contrast to riociguat, the NO donors failed to trigger SEAP expression in the transfected HEK-293 cells within 24 h (Fig. 2a), suggesting that this basic system is not sufficiently sensitive to NO.

Therefore, we next co-transfected cGMP-dependent PKG1 ($P_{hCMV}$-PKG1-pA$_{bGH}$)[37] and/or PKG2 ($P_{hCMV}$-PKG2-pA$_{bGH}$)[38] together with the components used in the first cycle. In the PKG1-expressing system, but not the PKG2-expressing system, the NO donors DETA NONOate and SNAP induced substantial levels of SEAP, indicating that co-expression of PKG1 promotes sGC-mediated NO sensing and amplifies the resulting cGMP production. Thus, inclusion of PKG1 was essential for hNORM functionality (Fig. 2b). The device thus developed showed potent and robust NO inducibility of SEAP

expression in a variety of human (A549, HEK-293, HeLa, HepG2 and HT-1080) and rodent (CHO-K1 and BHK-21) cell lines of different tissue origins, in addition to primary human-derived dermal fibroblasts (HDFn) (Fig. 2c). Since HEK-293 cells showed the best result, we next set out to construct transgenic HEK-293 cells stably expressing the above components.

## Generation and characterization of a stable hNORM-transgenic human cell line

To further characterize the performance of hNORM and to evaluate its potential for in vivo applications, we generated a stable monoclonal hNORM-transgenic HEK-293 cell line, HEK$_{hNORM1}$, which constitutively and stably co-expresses sGC, PKG1 and P$_{CRE}$-driven nanoluciferase (NLuc). We introduced NLuc instead of SEAP at this point because it is more sensitive than SEAP and enables more precise transgene expression profiling (Extended Data Fig. 1). We chose the Sleeping Beauty transposase (SB) system to achieve efficient stable integration (Methods). Thus, HEK-293 cells were co-transfected with engineered SB transposons containing the hNORM components sGC (pMMH185; SB$_{ITR}$-P$_{hPGK}$-sGCα-P2A-sGCβ-pA$_{bGH}$;P$_{SV40}$-iRFP670-P2A-Hygro-pA$_{p9}$-SB$_{ITR}$) and PKG1 (pMMH188; SB$_{ITR}$-P$_{hCMV}$-PKG1-pA$_{bGH}$;P$_{hCMV}$-mRuby2-P2A-Zeo-pA$_{p9}$-SB$_{ITR}$) as well as P$_{CRE}$-driven NLuc (pMMH186, SB$_{ITR}$-P$_{CRE}$-NLuc-pA$_{bGH}$;P$_{hCMV}$-YPet-P2A-Puro-pA$_{p9}$-SB$_{ITR}$). The resulting monoclonal cell line HEK$_{hNORM1}$ showed very low leakiness and robust induction of the transgene (135-fold with a half maximal effective concentration ~75 μM) (Fig. 3a) in a cGMP-dependent manner (Fig. 3b). When we examined the hNORM onset kinetics, we noticed that prolonged exposure to DETA NONOate is required to achieve marked transgene expression (Fig. 3c). This was further confirmed by real-time microscopy analysis using the P$_{CRE}$-driven fluorescent protein TurboGFP (Supplementary Videos 1–4). Transgene expression showed good reversibility at both the protein (Fig. 3d) and messenger RNA levels of the protein of interest (Extended Data Fig. 2). In addition, HEK$_{hNORM1}$ cells maintained high viability (Fig. 3e) at regulation-effective concentrations of the NO donor DETA NONOate, without marked cytotoxicity (Fig. 3f). Furthermore, the sGC-inhibitor drug methylene blue[39,40], clinically licensed for the treatment of methemoglobinemia[41], dose dependently repressed NO-triggered hNORM-mediated transgene expression, thereby providing adjustable dual-drug input control as well as representing a safety switch that could be used to reset hNORM in emergency situations (Fig. 3g).

## Mitochondrial ALDH2 increases NO-triggered hNORM performance in vivo

Next, we examined whether topical NG patches, which enable a continuous release of a specific amount of NG over a defined period of time, could be used to achieve convenient, precise and sustained production of NO, in place of the NO donors. To validate NO-mediated in vivo hNORM control by percutaneous delivery of NG, we subcutaneously implanted HEK$_{hNORM1}$ microencapsulated in coherent alginate–PLL–alginate capsules, which provide vascular connection and immuno-protection[7,42,43], and placed clinically licensed patches containing NG above the implant site (Fig. 4a). Indeed, topical NG patches (Deponit 10) of different sizes and NO content enabled precise control of the production and systemic delivery of target proteins by the subcutaneously implanted microencapsulated hNORM-transgenic cells (Fig. 4b).

Since biotransformation of percutaneously delivered NG to the trigger compound NO is dependent on ALDH2 (refs. 16,17), this reaction may be a rate-limiting step impacting the overall performance of hNORM in vivo. To ensure efficient in situ enzymatic conversion of NG into NO, we therefore engineered HEK$_{hNORM1}$ for additional constitutive expression of ALDH2. We transfected HEK$_{hNORM1}$ with pMMH187 (SB$_{ITR}$-P$_{hCMV}$-ALDH2-pA$_{bGH}$;P$_{hCMV}$-mTagBFP2-P2A-Blast-pA$_{p9}$-SB$_{ITR}$) and selected the ALDH2-transgenic monoclonal HEK$_{hNORM1}$-derived cell line HEK$_{hNORM2}$. Western blot and quantitative PCR (qPCR) analysis confirmed ectopic ALDH2 expression in HEK$_{hNORM2}$ (Fig. 4c and Extended Data Fig. 3a). The use of HEK$_{hNORM2}$ in place of HEK$_{hNORM1}$ increased the induction fold of NLuc in the hNORM system by over three-fold in mice treated with clinically licensed NG patches (130 μg per 24 h), indicating that ALDH2 engineering effectively boosts the NG sensitivity and the overall in vivo performance of hNORM (Fig. 4d). This was confirmed by quantification of NO in skin tissue of mice implanted with encapsulated HEK$_{hNORM1}$ or HEK$_{hNORM2}$ following transdermal NG-patch treatment (Fig. 4e).

## Reversibility and local nature of hNORM control in mice

NG has a short half-life of about 5 min, and therefore we expected that placement of the NG-delivering skin patch on the skin surface as close as possible to the implant site (immediately above it) would minimize the effective dose required for hNORM control. This would also result in lower systemic NG levels, which might both increase the specificity and reduce pleiotropic effects in clinical implementations of hNORM. NO itself, in situ generated in the implanted cells, has a half-life of only a few seconds, and its action would be local. Indeed, mice in which NG patches were placed away from the subcutaneous implant site or NG-patch-treated mice with intraperitoneal HEK$_{hNORM2}$ implants did not show detectable systemic levels of NLuc, confirming hNORM's specific short-distance responsiveness to the NG patches (Fig. 4f), without substantially influencing deep tissues.

These results suggest that hNORM should be easily tunable and also that its action could be quickly stopped in an emergency situation simply by removing the NG transdermal patch. To confirm the in vivo reversibility, we subcutaneously implanted mice with microencapsulated HEK$_{hNORM2}$ and sequentially applied and removed topical NG patches at 24 h intervals while profiling the blood NLuc levels (Fig. 4g). The blood level of NLuc reproducibly reached a maximum within 24 h after administration of the NG patch and decreased after removal of the patch for six consecutive cycles of application and removal, indicating good reversibility, robustness and reliability of hNORM control in vivo.

---

**Fig. 3 | hNORM shows high efficacy and tunability, and can be blocked by methylene blue. a**, Dose dependence of the HEK$_{hNORM1}$ monoclonal cell line. HEK$_{hNORM1}$ cells (5 × 10$^4$ cells per well) were treated with different concentrations of DETA NONOate, and the NLuc levels (shown as relative luminescence unit (RLU)) in the supernatants were quantified 24 h thereafter. Data are presented as mean ± s.d. ($n$ = 4), and $P$ values were calculated using a two-tailed, unpaired Student's $t$-test. **b**, Intracellular cGMP in HEK$_{hNORM1}$ described in the previous experiment was quantified by a competitive ELISA. Data are presented as mean ± s.d. ($n$ = 4), and $P$ values were calculated using a two-tailed, unpaired Student's $t$-test. **c**, Induction kinetics in mammalian cells. HEK$_{hNORM1}$ cells (5 × 10$^4$ cells per well) were cultured in medium supplemented with vehicle (sterile water) or DETA NONOate (100 μM). Samples from the supernatants were collected for NLuc quantification every 4 h for 28 h. Data are presented as mean ± s.d. ($n$ = 4), and $P$ values were calculated using a two-tailed, multiple unpaired Student's

$t$-test. **d**, Evaluation of reversibility in vitro. HEK$_{hNORM1}$ cells were cultured at day 0 and DETA NONOate was added at days 1, 3 and 5, while drug wash-out was conducted at days 2, 4 and 6. Samples were taken every day at the same time, just before drug addition and removal. Data are presented as mean ± s.d. ($n$ = 4). **e**,**f**, Viability (**e**) and cytotoxicity (**f**) assessment of HEK$_{hNORM1}$ cells treated with DETA NONOate (100 μM) as described in **d**. Data are presented as mean ± s.d. ($n$ = 4). **g**, Inhibitory effect of methylene blue on transgene expression. HEK$_{hNORM1}$ cells were seeded at a density of 5 × 10$^4$ cells per well. After 12 h, cells were treated with various concentrations of methylene blue in the presence and absence of DETA NONOate (100 μM). NLuc levels in the supernatants were analysed 24 h thereafter. Data are presented as mean ± s.d. ($n$ = 4), and $P$ values were calculated using a two-tailed, unpaired Student's $t$-test. All data shown are biological replicates. Source data are provided as a Source Data file.

## NG-patch-controlled GLP-1 expression restores blood glucose homeostasis in experimental type 2 diabetes without cardiovascular interference

For proof of concept, we next focused on the delivery of glucagon-like peptide-1 (GLP-1) in a mouse model of type 2 diabetes, which is one of the dynamically most challenging metabolic disorders. GLP-1 is a

metabolic peptide hormone that decreases blood glucose by stimulating insulin secretion from pancreatic beta cells and inhibiting glucagon secretion from pancreatic alpha cells[44]. Long-lasting GLP-1 analogues, including dulaglutide (Trulicity), exenatide (Byetta), liraglutide (Victoza) and semaglutide (Ozempic, Wegovy and Rybelsus), have recently come into the limelight as blockbuster drugs for the treatment

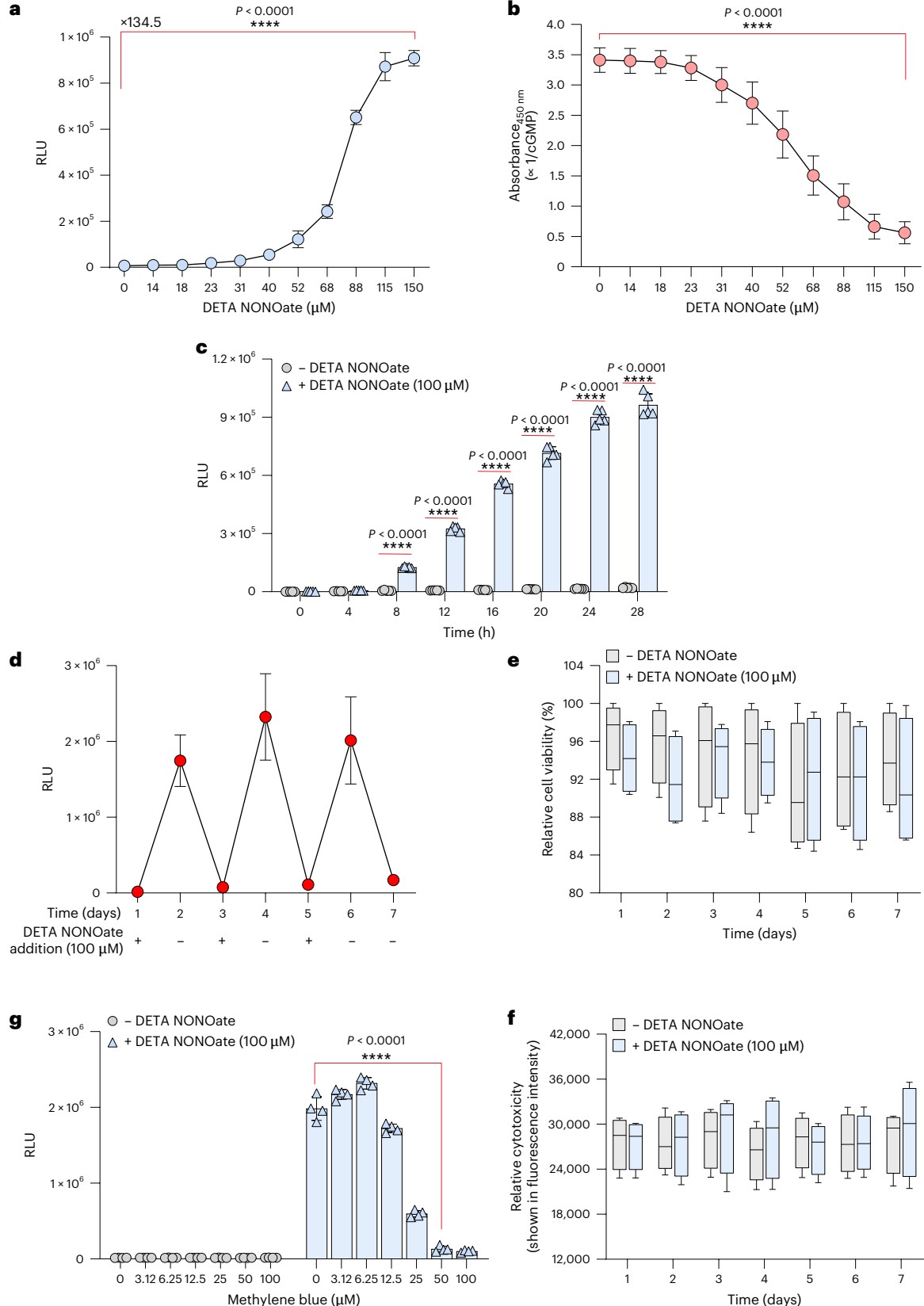

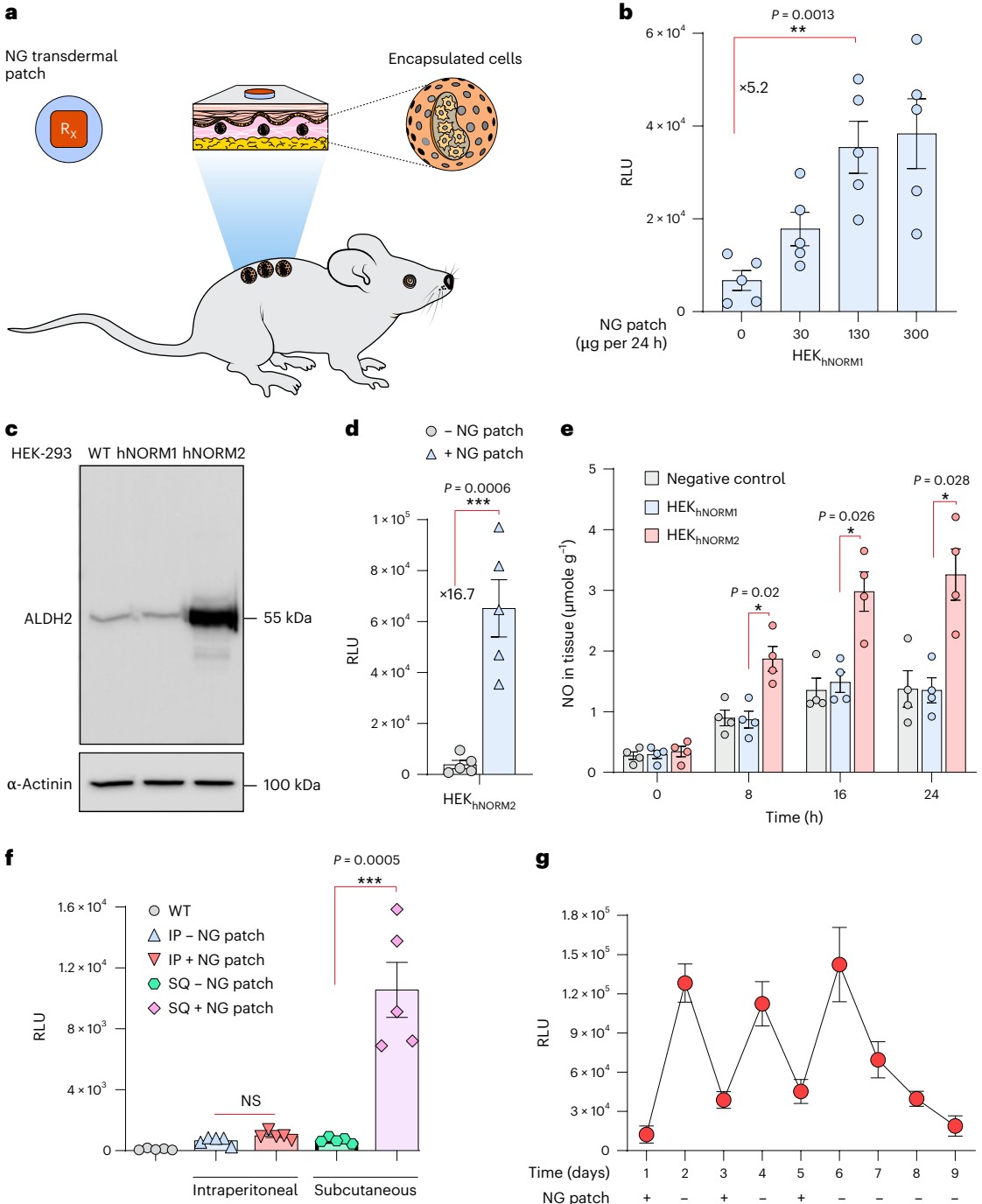

**Fig. 4 | Mitochondrial ALDH2 increases NO-triggered NLuc release upon induction by topical NG patches. a**, Schematic illustration of the in vivo experiment conducted to examine the performance of engineered cells in mice. **b**, Alginate–PLL–alginate encapsulated HEK$_{hNORM1}$ cells ($5 \times 10^6$ cells in total) were first implanted subcutaneously into C57BL/6 mice (Methods). At 24 h after device implantation, different dosages of NG patches were topically applied above the cell implant. NLuc levels in the blood were measured 24 h following the drug administration. Data are presented as mean ± s.e.m. ($n = 5$), and $P$ values were calculated using a two-tailed, unpaired Student's $t$-test. **c**, Immunoblotting for ALDH2 of parental HEK-293 WT, HEK$_{hNORM1}$ and HEK$_{hNORM2}$. α-Actinin was used as a loading control. **d**, NLuc levels in blood of mice as described in **b** using HEK$_{hNORM2}$ designer cells instead of HEK$_{hNORM1}$, and an NG-patch dose of 130 µg per 24 h. Data are presented as mean ± s.e.m. ($n = 5$), and $P$ values were calculated using a two-tailed, unpaired Student's $t$-test. **e**, NO levels in skin tissue of mice following the administration of an NG patch. Encapsulated HEK-293 WT, HEK$_{hNORM1}$ or HEK$_{hNORM2}$ cells were subcutaneousinfluencing deep tissues.ly delivered into C57BL/6 mice, and an NG patch (130 µg per 24 h) was applied on top of the implantation site as

described in **b**. NO levels in skin tissue beneath the NG patches were assessed at 8 h intervals. Data are presented as mean ± s.e.m. ($n = 4$). Statistical significance was analysed by two-way ANOVA and $P$ values were calculated using Tukey's multi-comparison tests. **f**, HEK$_{hNORM2}$-containing microcapsules were implanted either subcutaneously or intraperitoneally into four groups of C57BL/6 mice ($5 \times 10^6$ cells per mouse). At 2 h following the implantation, NG patches (130 µg per 24 h) were applied to the back and NLuc in the blood was quantified after 24 h. WT mice implanted with parental HEK-293 cells were used as a negative control. Data are presented as mean ± s.e.m. ($n = 5$), and $P$ values were calculated using a two-tailed, unpaired Student's $t$-test. IP, intraperitoneal; SQ, subcutaneous; NS, not significant. **g**, In vivo evaluation of reversibility using NG patches. HEK$_{hNORM2}$-containing microcapsules were subcutaneously implanted into mice at day 0 using $5 \times 10^6$ cells per mouse. NG patches (130 µg per 24 h) were applied above the implants on days 1, 3 and 5 (for 24 h each time), while on days 2, 4, 6, 7, 8 and 9, mice were kept without an NG patch. Blood samples were taken every day at the same time just before patch application and removal. Data are presented as mean ± s.e.m. ($n = 4$). All data shown are biological replicates. Source data are provided as a Source Data file.

of type 2 diabetes and associated obesity as well as lifestyle drugs to reduce body weight in the absence of exercise by increasing satiety and lowering gastric emptying[45].

For this experiment, we generated a stably transgenic monoclonal NG-patch-triggered NO-inducible GLP-1-producing cell line, HEK$_{hNORM3}$, by co-transfecting sGC (pMMH185; SB$_{ITR}$-P$_{hPGK}$-sGCα-P2A-sGCβ-pA$_{bGH}$: P$_{SV40}$-iRFP670-P2A-Hygro-pA$_{p9}$-SB$_{ITR}$), PKG1 (pMMH188; SB$_{ITR}$-P$_{hCMV}$-PKG1-pA$_{bGH}$:P$_{hCMV}$-mRuby2-P2A-Zeo-pA$_{p9}$-SB$_{ITR}$), ALDH2 (pMMH187; SB$_{ITR}$-P$_{hCMV}$-ALDH2-pA$_{bGH}$:P$_{hCMV}$-mTagBFP2-P2A-Blast-pA$_{p9}$-SB$_{ITR}$) and a P$_{CRE}$-driven GLP-1 variant (pMMH213; SB$_{ITR}$-P$_{CRE}$-GLP-1-Fc-P2A-N Luc-pA$_{bGH}$:P$_{hCMV}$-YPet-P2A-Puro-pA$_{p9}$-SB$_{ITR}$). The selected monoclonal HEK$_{hNORM3}$ showed almost 40-fold GLP-1 induction when triggered by the NO donor DETA NONOate in vitro (Fig. 5a and Extended Data Fig. 3b). To validate that NO does not influence GLP-1 bioactivity, we evaluated the pharmacological activity of GLP-1 produced by HEK$_{hNORM3}$ in response to NO by using the custom-engineered GLP-1 sensor cell line HEK$_{GLP1R}$. In HEK$_{GLP1R}$ cells, ectopic expression of human GLP-1 receptor (pGLP1R; P$_{hCMV}$-GLP1R-pA$_{bGH}$[46]) is functionally rewired to the expression of SEAP (pCK53, P$_{CRE}$-SEAP-pA$_{bGH}$) via a synthetic signalling cascade, enabling us to profile bioactive GLP-1 in the culture supernatant of HEK$_{hNORM3}$ cells (Extended Data Fig. 4).

We then implanted microencapsulated HEK$_{hNORM3}$ cells subcutaneously into type 2 diabetic mice and confirmed that the NG patch induces GLP-1 secretion in a dose-dependent manner (Extended Data Fig. 5). Similar to the results of the previous experiment (Fig. 4b), the NG patch at 130 μg per 24 h was sufficient to induce GLP-1 secretion. Following dosage validation, HEK$_{hNORM3}$-containing microcapsules were subcutaneously implanted in db/db mice and a transdermal NG-delivery patch (Deponit 10; 130 μg NG per 24 h) was applied on the skin above the implant site. The patch was replaced every other day, and the systemic active GLP-1 and blood glucose levels were profiled for 35 days. Diabetic mice treated with NG patches showed substantially higher blood GLP-1 levels than the untreated group (Fig. 5b), and normoglycaemia was restored throughout the entire treatment period of 35 days (Fig. 5c). Control mice with parenteral cell (non-engineered HEK-293 WT cells) implants showed similar levels of both GLP-1 and glucose with or without NG patches, confirming that the therapeutic effect stems from the stimulated designer cells and not from the patch itself (Extended Data Fig. 6). Profiling of the mean arterial blood pressure (MAP) (Fig. 5d), as well as the heart rate (Fig. 5e), of treated mice confirmed the absence of cardiovascular side effects over the entire treatment period. The absence of NG-associated cardiovascular effects indicates that the amount of NG delivered from the NG patches into the blood circulation in mice harbouring HEK$_{hNORM3}$ cell implants is insufficient to cause cardiovascular changes, despite the delivery of sufficient NG to activate hNORM. Since the used dosage of transdermal NG (130 μg per 24 h) is enough to reduce blood pressure[47], we hypothesized that HEK$_{hNORM3}$ cells that overexpress ALDH2 form a metabolic barrier, mediating efficient biotransformation of NG inside the subcutaneous cell implant before it can reach the blood circulation. To test this hypothesis, encapsulated cell implants containing either non-engineered HEK-293 WT or HEK$_{hNORM3}$ cells were subcutaneously implanted in WT mice. After 24 h, NG patches (130 μg per 24 h) were applied directly above the implants, and the cardiovascular parameters were monitored. A reduction in MAP (Fig. 5f) and an increase in heart rate (Fig. 5g) were observed only in mice harbouring HEK-293 WT cell implants, but not in mice with HEK$_{hNORM3}$ cell implants, supporting the idea that the HEK$_{hNORM3}$ cells intercept NG before it can reach the systemic circulation.

### Percutaneous NG-patch-controlled GLP-1 release by implanted HEK$_{hNORM3}$ cells reduces insulin levels, attenuates insulin resistance, resets body weight and restores glucose homeostasis of treated mice

Dysregulated and chronically elevated blood glucose levels, insulin resistance and obesity are hallmarks of type 2 diabetes, triggering a panoply of secondary medical conditions such as cardiovascular disorders, eye, kidney, and skin damage as well as dementia. Therefore, we next profiled the impact of NG-patch-controlled GLP-1 expression on experimental type 2 diabetes by treating type 2 diabetic mice bearing subcutaneously implanted HEK$_{hNORM3}$ with clinically licensed NG patches for 35 days. The results demonstrated that hNORM normalized systemic insulin levels (Fig. 6a), reset the body weight to a normal level (Fig. 6b), restored long-term blood glucose homeostasis (Fig. 6c) and eliminated insulin resistance (Fig. 6d). In addition, glucose tolerance (Fig. 6e) and insulin tolerance (Fig. 6f) tests after 35 days of patch-controlled hNORM treatment confirmed the reversal of chronic insulin resistance as well as hyperglycaemia, and the treated animals showed normal postprandial glucose-control and insulin-response dynamics (Fig. 6e,f). These results were further validated through the continuous measurement of glucose levels (Fig. 6g), MAP (Extended Data Fig. 7a) and heart rate (Extended Data Fig. 7b) for 24 h in mice following 35 days of hNORM treatment.

## Discussion

Over four decades after the approval of the first biopharmaceutical product, recombinant human insulin, the limitations of biopharmaceuticals, such as the need for downstream processing, formulation and testing in manufacturing, as well as the requirement for administration of the recombinant protein products at regular intervals, have become clear[48–50]. Consequently, there is a gathering momentum for the introduction of cell-based therapies using remotely controlled designer cells engineered for in situ biopharmaceutical production as the next pillar of medicine. This is well exemplified by CAR-T cell-based therapy targeting B-cell acute lymphoblastic leukaemia[51]. However, serious side effects such as cytokine release syndrome have highlighted the need for extremely precise control of cellular behaviour in cell-based therapies[52]. Remote control modalities are also required for in situ-produced biologics to adapt their delivery to changing dosing regimens, to titrate their release to the required therapeutic window and to halt the therapeutic intervention if necessary. Currently available transgene-control modalities typically incorporate gene switches inducible by a wide range of triggers, including antibiotics[1], vitamins[53], food additives[3], cosmetics[4], volatile fragrances[5], light[6], electricity[7], radio waves[8] and heat[54]. However, restricted bioavailability, pleiotropic side effects, functional interference and unfavourable pharmacodynamics may jeopardize the control performance of systemically administered trigger compounds[55], while traceless control modalities may require high energy input, may involve unphysiological chemical or inorganic co-factors with side effects, poor bioavailability or short half-lives, may suffer from illumination-based cytotoxicity or may be confounded by any fever-associated medical condition[56].

In this context, transdermal delivery of trigger compounds by topical administration appears to be a promising solution for controlling the cellular behaviour of subcutaneously implanted designer cells because it enables local remote control, which may increase specificity and sensitivity while reducing adverse effects and the required trigger dose, compared with systemic delivery routes[57]. Also, topical administration can be easily achieved by means of cream and patch formulations, which should increase patient convenience and compliance and reduce the risk of acute dosing errors[58]. Indeed, dosing and medication adherence remain major concerns in the long-term management of chronic disorders[59,60]. In recent years, increasing efforts have been made to overcome these challenges. For example, a transdermal polymer-based microneedle patch that releases pre-loaded insulin as a function of high glucose levels was successfully developed and showed promising results in minipigs[61]. This technology along with cell-based approaches, such as hNORM, is expected to advance percutaneous control modalities for biopharmaceutical delivery. In our work, the selection of GLP-1 for validation of our NG/NO-tunable gene switch is strategic. GLP-1 and its derivatives, including the blockbuster

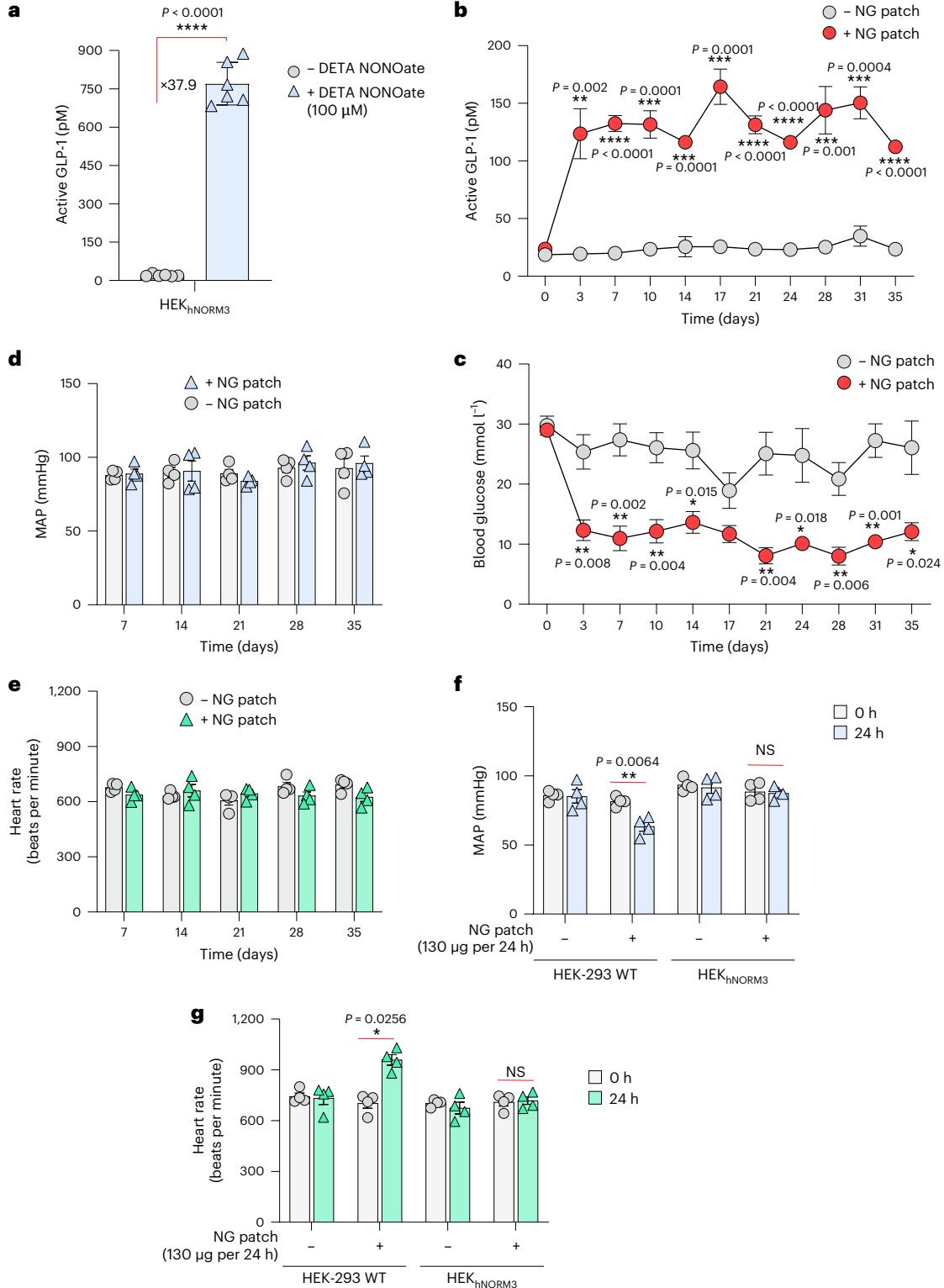

**Fig. 5 | hNORM-regulated GLP-1 secretion effectively reduces glucose levels in type 2 diabetic mice without affecting blood pressure or heart rate.** **a**, ELISA quantification of active GLP-1 levels in the supernatants of HEK$_{hNORM3}$ cells following DETA NONOate (100 μM) treatment for 24 h. Data are presented as mean ± s.d. ($n = 6$), and $P$ values were calculated using a two-tailed, unpaired Student's $t$-test. **b,c**, Long-term performance and anti-glycaemic effect of hNORM in db/db diabetic mice. HEK$_{hNORM3}$ cell-containing microcapsules ($5 \times 10^6$ cells per mouse) were subcutaneously implanted into db/db mice at days 0 and 14, and NG patches were topically applied in the vicinity of the implant. NG patches (130 μg per 24 h) were applied once every 2 days starting from day 0 until the end of the experiment. Plasma GLP-1 (**b**) and fasting glucose (**c**) levels were

analysed every 3 days for 35 days. Data are presented as mean ± s.e.m. ($n = 4$), and $P$ values were calculated using a two-tailed, unpaired Student's $t$-test. **d,e**, MAP (**d**) and heart rate (**e**) of mice described in the previous experiment measured at 7 day intervals. **f,g**, MAP (**f**) and heart rate (**g**) of WT mice harbouring subcutaneous cell implants that contain either HEK-293 WT or HEK$_{hNORM3}$ cells. NG patches (130 μg per 24 h) were applied at 24 h after the implantation, and cardiovascular parameters were measured at times 0 h and 24 h following the patch application. Data are presented as mean ± s.e.m. ($n = 4$). Statistical significance was analysed by two-way ANOVA, and $P$ values were calculated using Tukey's multi-comparison tests. All data shown are biological replicates. Source data are provided as a Source Data file.

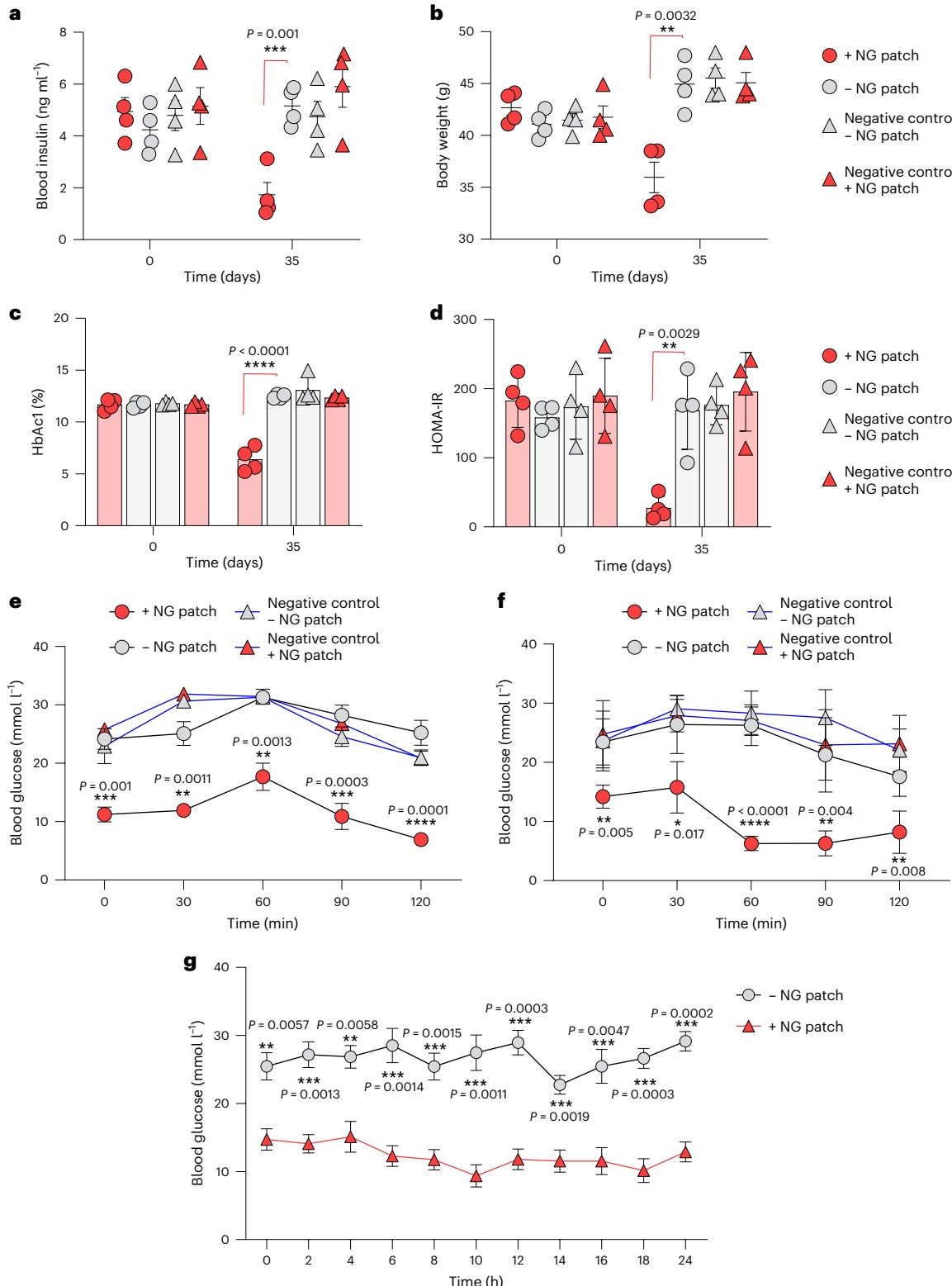

**Fig. 6 | Long-term hNORM-regulated GLP-1 delivery substantially ameliorates type 2 diabetes mellitus-associated metabolic changes in db/db diabetic mice.** **a**–**d**, Blood levels of insulin (**a**), HbAc1 (**b**), HOMA-IR (**c**) and body weight (**d**) of db/db mice treated with hNORM. db/db mice were subcutaneously injected with alginate–PLL–alginate encapsulated parental WT (negative control) or HEK$_{hNORM3}$ cells (5 × 10$^6$ cells per mouse) both at day 0 and day 14 as described in Fig. 4b. The treated groups received NG patches (130 µg per 24 h) once every 2 days for 35 days. Data are presented as mean ± s.e.m. (*n* = 4), and *P* values were calculated using multiple, unpaired Student's *t*-tests. **e**,**f**, Glucose tolerance test (**e**) and insulin tolerance test (**f**) of db/db mice (the same mice described previously) after 35 days of hNORM treatment. Data are presented as mean ± s.e.m. (*n* = 4), and *P* values were calculated using multiple, unpaired Student's *t*-tests. **g**, Glucose levels of db/db mice after 35 days of hNORM treatment as described above were monitored continuously for 24 h. Data are presented as mean ± s.e.m. (*n* = 3), and *P* values were calculated using multiple, unpaired Student's *t*-tests. All data shown are biological replicates. Source data are provided as a Source Data file.

drugs Ozempic, Saxenda and Wegovy, are pivotal in managing type 2 diabetes and obesity. These conditions have a global impact, affecting over 10% of the population today, as reported by the International Diabetes Federation. Moreover, their prevalence is expected to rise, with nearly one-quarter of the human population projected to be affected by 2035, according to the World Obesity Atlas 2023. Furthermore, this technology could also potentially be used for the management of other chronic diseases where protein therapeutics are indicated. In situ bioproduction of recombinant erythropoietin in patients with chronic kidney failure, delivery of monoclonal antibodies to treat chronic immune diseases and enzyme replacement therapy are just a few of the possibilities[62,63].

Gene switches responsive to licensed cosmetics additives, such as the penetration enhancer phloretin[64], the preservative paraben[4], the soothing compound menthol[65] and the tonic fragrance spearmint[66], have all been used to percutaneously remote-control transgene expression in subcutaneous implants following topical application. However, phloretin can be associated with cell toxicity[67], parabens are suspected of perturbing the hormone system and causing cancer[68], and menthol and spearmint are difficult to dose and have rather long half-lives, which compromise the reversibility of the gene switches[69]. By contrast, NG formulated in skin patches for topical percutaneous delivery has a long track record as a safe, efficient and well-tolerated drug, having been in clinical use for over 130 years, and therefore we consider it a superior trigger compound with better expression kinetics (Supplementary Fig. 1).

Local conversion of NG to NO for activation of hNORM is catalysed by ALDH2, which is present in somatic cells. However, we found that ectopic expression of ALDH2 in NO-responsive designer cells markedly increases NG conversion. Thus, the introduction of ALDH2, as well as PKG1, which amplifies the cGMP signal, increases the specificity of hNORM, reduces the required trigger compound dose and minimizes potential pleiotropic effects around the implant site. While overexpressing ALDH2 enhances hNORM performance in vivo, it not obligatory to express it in the designer cells. This can be inferred from Fig. 4b and also from the results of co-administration of the ALDH2 inhibitor disulfiram[70] and ALDH2 activator Alda-1 (ref. 71) (Supplementary Fig. 2). Therefore, the likelihood of interference resulting from the co-administration of commonly prescribed drugs seems small (Supplementary Fig. 2).

A key feature of hNORM is that it is entirely composed of human control components, which should minimize the risk of immune responses, enable seamless integration into the cellular metabolism and ensure physiologically compatible control dynamics. This was confirmed by the absence of a systemic (Supplementary Fig. 3) or local (Supplementary Fig. 4) immune response, as judged from the results of cytokine profiling and immunohistochemistry analysis around the implant tissue, respectively. Furthermore, a thorough examination of the cell implant indicated good long-term stability of the device, as well as the absence of fibrosis (Supplementary Fig. 4). Crucially, as illustrated in Fig. 5b, we found that alginate-encapsulated hNORM-transgenic HEK-293 cells exhibited responsiveness to NO and consistently released therapeutic levels of GLP-1 throughout the entire 35-day study period. These results are in accordance with other studies in which alginate-microencapsulated cell implants have been used in vivo[7,65,72]. In addition, RNAseq analysis of the implanted cells and the host tissue surrounding the implant did not show a substantial change in RNA levels, suggesting the absence of off-target toxicity or interference with the host tissue (Supplementary Fig. 5).

The all-human hNORM could interfere with the endogenous NO-regulatory network upon stimulation with NG patches, in contrast to orthogonal gene switch modalities. However, direct in situ conversion of NG into NO inside the designer cells overexpressing the metabolizing enzyme ALDH2 and the short distance between the NG patch and implanted cells should minimize the effective dose required for hNORM regulation, with little impact on systemic NG levels. This feature, in combination with the extremely short-lived nature of NO, is expected to provide high specificity with few pleiotropic effects in clinical implementations of hNORM. Indeed, we confirmed that the operation of the hNORM system had no impact on key cardiovascular parameters such as heart rate and blood pressure over a 35-day experimental period (Fig. 5d–g). Furthermore, physiological fluctuations in endogenous NO signalling are not expected to interfere with hNORM activity, as judged from the steady levels of plasma GLP-1 in NG-untreated mice over 35 days (Fig. 5b). This behaviour is consistent with the fact that prolonged exposure (around 8 h) and relatively high levels of NO are required for substantial hNORM-mediated gene expression (Fig. 3c)–two criteria that can be met by external intervention[73,74].

Overall, we consider that hNORM has many advantages that may make it particularly suitable for applications in personalized cell-based therapies for acute and chronic medical conditions. For example, the convenient, non-invasive, precise and on-demand trigger delivery from commercially available and Food and Drug Administration (FDA)-approved NG patches is expected to enhance patients' compliance with treatment and to improve their quality of life, and should ease the path to industrial-scale production of cell-based therapies[75]. In addition, the low cost, ready availability and unsophisticated storage requirements of NG patches, in addition to long clinical experience with this drug, are in marked contrast to expensive GLP-1 analogues, which require repeated injections and well-regulated storage conditions.

Despite the clinical promise of hNORM, further work will be needed to translate this proof-of-concept study into human applications. For example, the source and nature of the cells intended for human use must be meticulously explored, as HEK-293 cells are not clinically licensed for humans. Another important point that should be investigated is the long-term stability of the implant in humans. While we observed impressive stability of the alginate–PLL–alginate microcapsules at 35 days post-implantation in mice, confirmation will be needed in humans owing to species differences in extracellular and degrading enzyme repertoires, as well as environmental factors. In this study, microencapsulated cells were implanted on the animal's back, a site that generally is not exposed to physical force or stressful contact with other objects found in cages. In human application, the implantation site should be carefully selected, and the stability of the implant must be assessed under conditions of normal movement and physical activities.

## Outlook

The future application of hNORM may offer a useful drug delivery method for targeting chronic medical conditions that are currently managed by an invasive and frequent injection-based regime. Using a delivery system based on human-derived components triggered on-demand by commercially available and clinically approved NG patches, hNORM technology can potentially enhance a patient's adherence to treatment and improve their overall quality of life. In addition, NG patches are relatively inexpensive and widely available and do not require a specialized storage condition. Therefore, hNORM may present a more affordable alternative technology to current injection-based therapies.

## Methods

### Drugs and chemicals

DETA NONOate (catalogue number ALX-430-014) and SNAP (catalogue number BML-CN210) were purchased from Enzo Life Sciences AG. Owing to the low stability of these NO donors in an aqueous solution, solutions were freshly prepared in sterile water and used immediately in each experiment. Riociguat (catalogue number 9000554) was purchased from Cayman Chemical. Menthol (catalogue number 63660), phloretin (catalogue number P7912), disulfiram (catalogue number PHR1690), Alda-1 (catalogue number SML0462), metformin (catalogue

number 317240), enalapril (catalogue number E6888), atorvastatin (catalogue number SML3030), atenolol (catalogue number A7655), aspirin (catalogue number A5376), methylene blue (catalogue number 122965-43-9) and recombinant GLP-1 (7-36) (catalogue number G9416) were purchased from Sigma-Aldrich (catalogue number 122965-43-9). Clinical NG transdermal patches (Deponit 10 mg per 24 h; size 18 cm$^2$) were obtained from CPS Cito Pharma Services (SKU 1029890). To obtain different dosages (30 μg, 130 μg and 300 μg per 24 h), NG patches of 6.25 mm$^2$, 25 mm$^2$ and 56.25 mm$^2$, respectively, were used.

## Plasmid construction
Gene expression vectors were constructed either by using restriction enzymes (New England Biolabs) followed by ligation with T4 DNA ligase (New England Biolabs, catalogue number M0202L) or by Gibson assembly (New England Biolabs, catalogue number E2611L). For restriction enzyme-based cloning, digested plasmid backbones were dephosphorylated with Antarctic phosphatase before ligation (New England Biolabs, catalogue number M0289L). PCR reactions were performed using Q5 High-Fidelity DNA polymerase (New England Biolabs, catalogue number M0491L). For Gibson assembly, the PCR products were amplified using primers having 15–20 bp complementary sequences to each end of the linearized vector. Detailed information about plasmid cloning is presented in Supplementary Table 1. DNA sequence of the cloned genes in this study are presented in Supplementary Table 3. Plasmids were transformed and propagated in XL10-Gold ultra-competent *Escherichia coli* (New England Biolabs, catalogue number C2992) and extracted using a plasmid miniprep kit (Zymo Research, catalogue number D4054) or a ZymoPURE II plasmid midiprep kit (Zymo Research, catalogue number D4200).

## Cell culture
Human embryonic kidney cells (HEK-293T, ATCC CRL-11268), HeLa cells (HeLa, ATCC CCL-2), HepG2 cells (HEPG2, ATCC HB-8065), baby hamster kidney cells (BHK, ATCC CCL-10) and HT-1080 cells (HT1080, ATCC CCL-121) were cultivated in Dulbecco's modified Eagle's medium (DMEM; ThermoFisher Scientific, catalogue number 10566016). Chinese hamster ovary cells (CHO-K1, ATCC CCL-61) and A549 cells (A549, ATCC CCL-185) were cultivated in Ham's F-12K (Kaighn's) medium (Gibco F-12K, catalogue number 21127-022, ThermoFisher Scientific). Primary dermal fibroblasts (normal, human, neonatal) (HDFn; ATCC PCS-201-010) were cultured using Fibroblast Basal Medium (ATCC PCS-201-030) and Fibroblast Growth Kit-Low Serum (ATCC PCS-201-041) according to the supplier's instructions. DMEM and Ham's F-12K (Kaighn's) medium were supplemented with 10% fetal bovine serum (Sigma-Aldrich, catalogue number F7524, lot number 022M3395) and penicillin (100 U)–streptomycin (100 μg) solution (Sigma-Aldrich, catalogue number P433), and cultured under a humidified atmosphere of 5% $CO_2$ in air at 37 °C. Passaging of pre-confluent cultures was performed by trypsinization with 0.05% trypsin-EDTA (Life Technologies, catalogue number 25300-054) for 5 min at 37 °C. Cells were transferred to 10 ml of cell culture medium and centrifuged for 1 min at 200 × *g*. The supernatant was discarded, and the cells were re-suspended in a fresh medium. Cell number and viability were quantified using CellDrop Automated Cell Counters DeNovix (Labgene Scientific).

## Transient transfection
For plasmid transfection in a 96-well format, cells were seeded at a density of 50,000 cells per 1 cm$^2$ in 100 μl medium for 24 h. For HEK-293 cells, 100 μl of serum and antibiotics-free minimum essential medium (MEM; ThermoFisher Scientific, catalogue number 11095080) containing a 1:5 DNA:PEI mixture (polyethyleneimine, MW 40,000; Polysciences, catalogue number 24765) with a total DNA amount of 350 ng cm$^{-2}$ was added dropwise to the cells, and the plate was incubated for 12 h. For the transfection of other cell lines, Lipofectamine 3000 Transfection Reagent (ThermoFisher Scientific, catalogue number L3000015) was used according to the manufacturer's instructions.

## SEAP quantification
SEAP levels in the cell culture medium were determined as follows: 20 μl of the culture supernatant was mixed with 80 μl ddH$_2$O and heat-inactivated for 30 min at 65 °C. Then, 80 μl of 2× SEAP buffer (20 mM homoarginine, 1 mM MgCl$_2$, 21% (v/v) diethanolamine, pH 9.8) and 20 μl of 120 mM *para*-nitrophenyl phosphate (Acros Organics, catalogue number 128860100) solution in 2× SEAP buffer were added to each well, and the absorbance at 405 nm was measured at 37 °C using a Tecan M1000 plate reader (Tecan Group). SEAP concentrations were calculated from a standard curve.

## NLuc quantification
Blood samples were collected in BD Microtainer blood collection tubes and left for 30 min at room temperature, and then centrifuged at 6,000 RCF for 5 min, and the plasma was transferred to new tubes. For in vitro experiments, samples were taken directly from the culture supernatants. For luminescence measurement, samples were transferred to black 384-well plates using 7.5 μl of the sample mixed with 7.5 μl working solution of the Nano-Glo Luciferase Assay System (Promega, catalogue number N1130) according to the manufacturer's instructions. A Tecan M1000 plate reader (Tecan Group) was used to measure luminescence.

## Stable cell line generation
For stable integration, genes of interest were cloned into Tier-3 plasmids: Tier3(SB)-YPet-Puro, Tier3(SB)-mTagBFP2-Blast, Tier3(SB)-mRuby2-Zeo and Tier3(SB)-iRFP670-Hygro. Sleeping Beauty transposase (pSB100x) was co-transfected for stable integration (Supplementary Table 1). Following their transfection, cells were treated with antibiotics for 3 days followed by sorting of florescence-positive cells using a BD FACS Aria Fusion cell sorter. After their recovery, cells were sorted again in a single-cell fashion directly into 96-well plates. Single cells were expanded, and the best-performing clones were selected for further study. Stable cell lines were validated by qPCR analysis of the transgene over 30 consecutive passages (Supplementary Fig. 6).

## Cell viability assay
Cell viability was evaluated using AlamarBlue Cell Viability Reagent (ThermoFisher, catalogue number DAL1025) according to the manufacturer's instructions.

## Cytotoxicity evaluation
Cytotoxicity under different conditions was evaluated using a CyQUANT Cytotoxicity Assay Kit (ThermoFisher Scientific, V23111) according to the manufacturer's instructions.

## Evaluating the pharmacological activity of secreted GLP-1 from HEK$_{hNORM3}$ cells following treatment with NO donor
HEK$_{hNORM3}$ cells (5 × 10$^4$ cells per well) were treated with DETA NONOate (100 μM) or vehicle (sterile water). After 24 h, the culture supernatants were transferred into active GLP-1 bio-detecting HEK-293 cells (HEK$_{GLP1R}$), which co-express GLP-1 receptor (P$_{hCMV}$-GLP1R-pA$_{bGH}$) and P$_{CRE}$-driven SEAP expression (pCK53, P$_{CRE}$-SEAP-pA$_{bGH}$). SEAP levels in the supernatants were measured after incubation for 24 h.

## Western blotting
Cells were collected, centrifuged at 1,000 × *g* for 5 min and washed twice in cold PBS. For cell lysis, RIPA buffer supplemented with protease and phosphatase inhibitor cocktail (ThermoFisher, catalogue number A32963) was added to the cell pellet and the mixture was vortexed for 20 min at 4 °C. Lysates were cleared by centrifugation at 12,000 × *g*

for 30 min at 4 °C. The 5× reduced Laemmli sample buffer was added, and the mixture was boiled for 5 min at 95 °C and loaded on SDS–PAGE. Protein quantification was performed using a Pierce BCA Protein Assay Kit (ThermoFisher, catalogue number 23227). Following SDS–PAGE, gels were blotted onto PVDF membranes using a Biorad PowerPac. Blots were blocked with 10% skim milk in TBST buffer at room temperature for 1 h. The following primary antibodies were used (all used at 1:1,000 dilution): monoclonal anti-ALDH2 antibody produced in rabbit (Cell Signaling Technology, catalogue number 18818) and mouse anti-α-actinin (Cell Signaling Technology, catalogue number 69758, clone E7U1O). Secondary HRP-conjugated goat anti-rabbit (catalogue number 111-035-144, polyclonal) and anti-mouse (catalogue number 115-035-003, polyclonal) antibodies were purchased from Jackson ImmunoResearch and used at a dilution of 1:10,000. PageRuler Plus Prestained Protein Ladder (10–250 kDa; ThermoFisher, catalogue number 26619) was used as a protein molecular weight marker set. Blots were developed using FUSION PulseTS (catalogue number 37480003, Vilber), and images were analysed using Adobe Illustrator software. All uncropped and unprocessed western blot images are provided in the Source Data file.

## RNA isolation and qPCR
Cultured cells were collected and washed twice with PBS. TRI Reagent (Sigma-Aldrich, catalogue number T9424) was added to the cell pellets and RNA was isolated according to the manufacturer's instructions. For each sample, complementary DNA was synthesized using a High-Capacity cDNA Reverse Transcription Kit (Applied Biosystems, catalogue number 4368814) using 2 µg of total RNA. For qPCR analysis, a 20 µl reaction mixture was prepared in 96-well plates as follows: 10 µl of SYBR Green Universal Master Mix (Applied Biosystems, catalogue number 4309155), 1 µl of diluted cDNA (1:3 diluted in nuclease-free water), 1 µl of 5 µM primer mix and 8 µl nuclease-free water. qPCR and data analysis were performed using QuantStudio 3 (ThermoFisher Scientific, catalogue number A28567). Primer pairs used for cDNA amplification are listed in Supplementary Table 2.

## RNAseq
Total RNA extraction was performed using a Quick-RNA miniprep kit (Zymo Research, catalogue number R1054). The quality of the isolated RNA was checked with a 4200 TapeStation device (Agilent). Human-derived engineered samples: RNAseq libraries were prepared using TruSeq Stranded mRNA reagents (Illumina) according to Illumina's guidelines. Input: 200 ng total RNA. Mouse surrounding tissues: RNAseq libraries were prepared using TruSeq Stranded Total RNA Library Prep Gold reagents (Illumina) according to Illumina's guidelines, but without performing the fragmentation step at 94 °C. Input: 150 ng total RNA. Quality control (QC) of sequencing libraries was performed using Fragment Analyzer systems (Advanced Analytical, AATI). Libraries were pooled and sequenced SR101 on an Illumina NovaSeq 6000 system using an SP flow cell and v1.5 reagents.

## Data analysis
Illumina sequencing data were demultiplexed and primary analysis was performed using a Snakemake workflow. The workflow includes trimmomatic (v0.35), alignment to the GRCh38 genome with hisat2 (v2.1.0), samtools (v1.9) to sort and index the alignment BAM files, and featureCounts from the Subread package (v2.0.1) to count reads in the gene ranges, using mouse Ensembl annotation v111. The count vectors for all samples were combined into a table, which was then subjected to secondary analysis in R. Quality control and sample consistency were checked with PCA using R package PCATools. The count table was processed in the secondary (statistical) analysis with R scripts using edgeR (v4.0.16). This resulted in lists of genes ranked for differential expression by *P* value. Benjamini–Hochberg adjusted *P* values were used to estimate the false discovery rate.

## Fluorescence microscopy
Images were taken by a Nikon Eclipse Ti2-E microscope (inverted). Light engine: Lumencor Spectra X (390/438/475/513/549/632 nm). Camera: Hamamatsu ORCA-Fusion. Objective: 10× Plan Apo with an NA of 4.5 (mrd00105). Videos were made using FIJI version 9.2.0.

## Blood pressure and heart rate measurement
Blood pressure and heart rate were analysed in the morning using a computerized, non-invasive tail-cuff system (Bioseb, catalogue number BP-2000).

## Cell encapsulation
Cell implants: Engineered HEK-293 stable cells were encapsulated in coherent alginate–poly-(L-lysine)–alginate beads (400 µm; 200 cells per capsule) using an Encapsulator (B395 Pro, Buechi Labortechnik AG) as described[42]. The following parameters were used: 20 ml syringe operated at a 20 ml min$^{-1}$ flow rate of 400 units, 200 µm nozzle with a vibration frequency of 1,200 Hz and a bead dispersion voltage of 1.5 kV. Mice were intraperitoneally injected with 1 ml of DMEM containing $5 \times 10^6$ encapsulated cells.

## Animal experiments
All experiments involving animals were performed either according to the directive of the European Community Council (2010/63/EU), approved by the French Republic (project number DR2013-v2), or according to the Animal Care Guidelines of the Ministry of Science and Technology of the People's Republic of China, approved by the Institutional Animal Care and Use Committee (IACUC) of Westlake University (protocol ID AP#24-088-XMQ). Twelve-week-old male C57BL/6 or db/db mice (BKS.Cg-Dock7m +/+ Leprdb/J, derived from C57BL/6J, Janvier Labs Saint-Berthevin) were used in this study. Animals were housed in a controlled room at 22 °C, 50% humidity, 12 h light–dark cycle with ad libitum access to standard diet and drinking water. Animals were randomly assigned to experimental groups. One day before implant delivery, the fur was locally removed from the back using commercial hair removal cream, Veet. NG patches were applied to the hairless area. Blood samples were taken from the tail or saphenous veins using a 20 µl glass micro-haematocrit capillaries (Avantor VWR, catalogue number 521-9100), transferred into blood collection tubes (BD Microtainer, catalogue number BDAM365968), incubated for 30 min at room temperature and centrifuged for 5 min at 6,000 RCF. The resulting serum samples were transferred to new 1.5 ml Eppendorf tubes and kept at −80 °C until analysis.

## Histological analysis
Histology and histopathological examination were performed at AnaPath Services GmbH. Two perpendicular sections from the test item implantation area (shaved area) were taken from each animal in the study. In addition, capsules were collected from the subcutaneous tissue under a stereoscopic microscope where appropriate. All samples were processed and embedded in paraffin wax, cut at an approximate thickness of 2–4 µm and stained with haematoxylin and eosin and Masson's trichrome. Apoptosis was evaluated by anti-caspase-3 immunohistochemistry (Cell Signaling Technology, lot number 47, catalogue number 9661S, in 1:500 dilution). The resulting slides were initially checked for quality and then examined under a light microscope by the study pathologist at AnaPath Services GmbH.

## Enzyme-linked immunosorbent assay (ELISA)
**Insulin quantification.** Mouse insulin (mINS) levels in serum were quantified using a mouse insulin ELISA kit according to the manufacturer's instructions (10-1247-01, Mercordia). Optical density was measured at 450 nm on a Tecan M1000 plate reader and the corresponding concentrations were calculated in Prism 9

(GraphPad Software) using a cubic-spline regression based on the measured absorbances of manufacturer-provided standard solutions.

**Active GLP-1 quantification.** GLP-1 levels in culture supernatants and active GLP-1 levels in mouse serum were quantified with the High-Sensitivity GLP-1 Active ELISA Kit (EZGLPHS-35K, Merck Millipore).

**HbA1c quantification.** HbA1c values were quantified with the Mouse Hemoglobin A1c Kit (80310, Crystal Chem).

**cGMP.** cGMP levels were quantified using a competitive enzyme-linked immunoassay Cyclic GMP XP Assay Kit (Cell Signaling Technology, catalogue number 4360).

### Glucose tolerance test
Fasted mice (12 h) received an intraperitoneal injection of aqueous 1.5 g kg$^{-1}$ D-glucose, and the glycaemic profile of each animal was tracked at 15 min, 30 min, 60 min, 90 min and 120 min thereafter.

### Insulin tolerance test
Fasted mice (4 h) received an intraperitoneal injection of 1 U kg$^{-1}$ recombinant insulin, and the glycaemic profile of each animal was tracked at 15 min, 30 min, 60 min, 90 min and 120 min thereafter.

### HOMA-IR calculation
The insulin resistance index was calculated according to the following formula: HOMA-IR = [fasting glucose (mmol l$^{-1}$) × fasting insulin (mU l$^{-1}$)]/22.5.

### Blood glucose measurement
Fasting blood glucose was measured after 4 h of food restriction using a clinically licensed glucometer (Accu-Check Instant, Roche).

### Cytokine profiling
Serum cytokine profiling in mice was performed using Mouse Cytokine Array Panel A (R&D Systems, catalogue number ARY006) according to the manufacturer's instructions.

### NO quantification in tissue
NO in tissue was quantified using a Nitric Oxide Assay Kit (Abcam, catalogue number 521-9100 ab272517) according to the manufacturer's instructions.

### Statistical analysis
The statistical significance of differences among groups was evaluated with a two-tailed, unpaired Student's *t*-test or analysis of variance (ANOVA) using GraphPad Prism. Differences are considered statistically significant at $P < 0.05$. The statistical test used and the significance are reported in the figures and figure legends.

### Inclusion and ethics
All animal experiments were conducted in compliance with the directive of the European Community Council (2010/63/EU) and approved by the French Republic (project number DR2013-v2, licenses G.C.-E.H., number 69266309, and S.X., number LTK4899). In addition, experiments were also performed following the Animal Care Guidelines of the Ministry of Science and Technology of the People's Republic of China and were approved by the Institutional Animal Care and Use Committee (IACUC) of Westlake University (protocol ID AP#24-088-XMQ). We adhered to rigorous ethical standards to ensure the welfare of the animals involved, and our study design was guided by the principles of the 3Rs: replacement, reduction and refinement.

### Reporting summary
Further information on research design is available in the Nature Portfolio Reporting Summary linked to this article.

## Data availability
The main data supporting the results in this study are available within the paper and its Supplementary Information. All raw and analysed data generated during the study are available from the corresponding author on reasonable request. Source data are provided with this paper.

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

## Acknowledgements

We thank M. Okoniewski for deep sequencing and statistical analyses as well as T. Gao for technical assistance. M.F. discloses support for the research described in this study from the European Research Council advanced grant ElectroGene (number 785800) and the National Centre of Competence in Research (NCCR) for Molecular Systems Engineering, the EC Horizon 2020 Framework Programme ENLIGHT (number 964497). M.X. discloses support for the publication of this study from the National Natural Science Foundation of China (NSFC Project 32071429). M.M. discloses support for the research described in this study from EMBO fellowship.

## Author contributions

M.M. and M.F. designed the project. M.M., S.X. and B.D. conducted the in vitro experiments. S.X., G.C.-E.H. and M.X. designed the animal experiments, and S.X. and G.C.-E.H. performed them. M.M., M.X. and M.F. analysed the data and wrote the paper.

## Funding

## Competing interests

The authors declare no competing interests.

## Additional information

**Extended data** is available for this paper at https://doi.org/10.1038/s41551-025-01350-7.

**Correspondence and requests for materials** should be addressed to Martin Fussenegger.

a

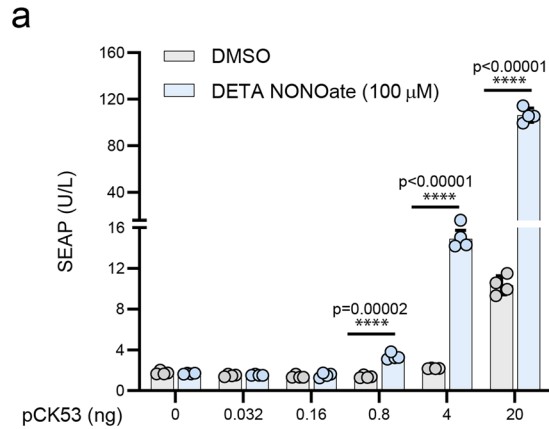

b

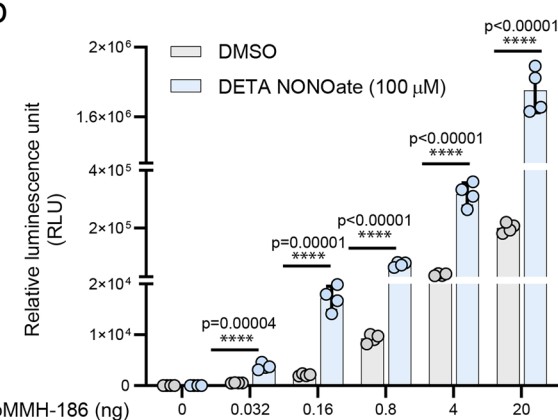

**Extended Data Fig. 1 | Evaluation of SEAP and NLuc reporter sensitivity under hNORM regulation.** HEK-293 cells were co-transfected with pMMH178 ($P_{hPGK}$-sGCα-pA$_{bGH}$), pMMH179 ($P_{hPGK}$-sGCβ-pA$_{bGH}$), PKG1β WT ($P_{hCMV}$-PKG1β-pA$_{bGH}$), along with different amounts of (**a**) $P_{CRE}$-driven SEAP reporter pCK53 ($P_{CRE}$-SEAP-pA$_{bGH}$) or (**b**) $P_{CRE}$-driven NLuc reporter pMMH186 (SB$_{ITR}$-$P_{CRE}$-NLuc-pA$_{bGH}$:$P_{hCMV}$-YPet-P2A-Puro-pA$_{p9}$-SB$_{ITR}$). At 24 h following the transfection, the

medium was changed to 100 µl of fresh medium containing either DMSO or DETA NONOate (100 µM). SEAP and NLuc levels in the supernatants were analyzed at 24 h after drug addition. Data are presented as means ± s.d., n = 4, and p values were calculated using a two-tailed, unpaired Student's t-test. Source data are provided as a Source Data file.

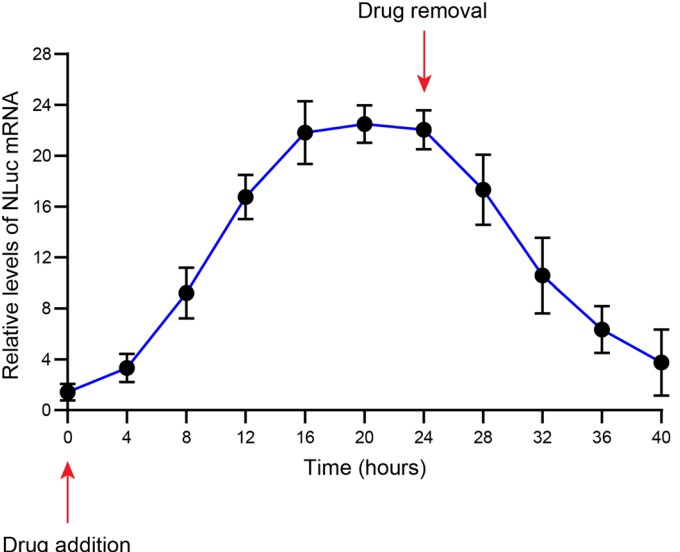

**Extended Data Fig. 2 | Expression kinetics of NLuc mRNA as a function of DETA NONOate presence.** HEK$_{hNORM1}$ cells were treated with DETA NONOate (100 µM) at the beginning of the experiment, and cells were harvested every 4 h. At 24 h, DETA NONOate was removed, and the medium was replaced with fresh medium. Cell samples were taken at the same time intervals until the end of the experiment (40 h). Total RNA was isolated and NLuc levels were quantified by qPCR. Values are shown as NLuc mRNA levels relative to corresponding untreated cells. Data are presented as means ± s.d., n = 3. Source data are provided as a Source Data file.

a

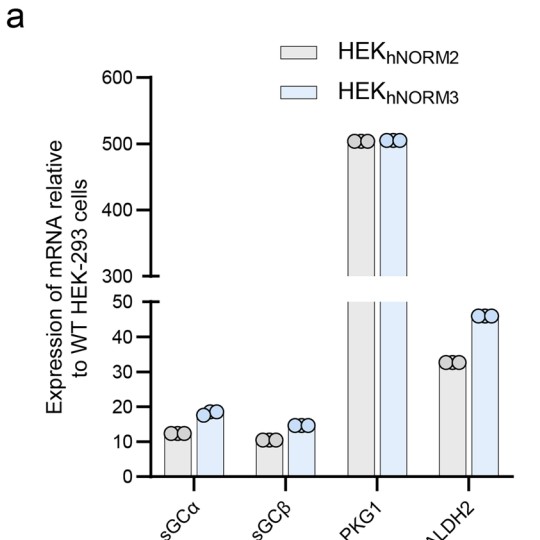
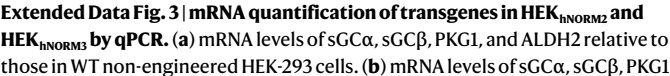

b

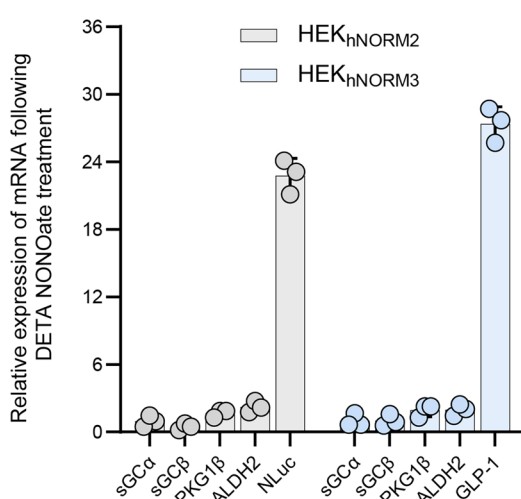

**Extended Data Fig. 3 | mRNA quantification of transgenes in HEK_hNORM2 and HEK_hNORM3 by qPCR.** (**a**) mRNA levels of sGCα, sGCβ, PKG1, and ALDH2 relative to those in WT non-engineered HEK-293 cells. (**b**) mRNA levels of sGCα, sGCβ, PKG1, ALDH2, NLuc, and GLP-1 following DETA NONOate (100 μM) treatment for 24 h. Values are shown relative to corresponding untreated cells. Data are presented as means ± s.d., n = 3. Source data are provided as a Source Data file.

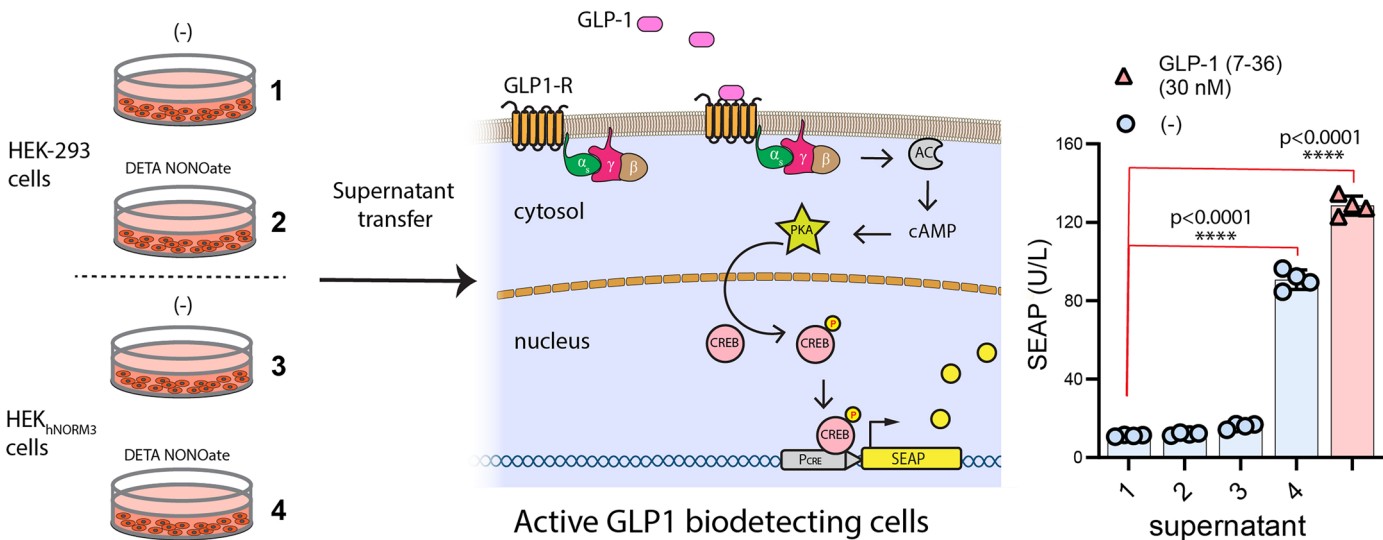

**Extended Data Fig. 4 | GLP-1 secreted via hNORM is pharmacologically active.** The indicated supernatants of parental HEK-293 or HEK$_{hNORM3}$ cells were harvested after 24 h treatment with DETA NONOate (100 µM) or vehicle (left). The culture supernatants (annotated 1-4) were transferred to active GLP-1 bio-detecting HEK-293 cells (HEK$_{GLP1R}$), which express GLP-1 receptor ($P_{hCMV}$-GLP1R-pA$_{bGH}$[3]) rewired to $P_{CRE}$-driven SEAP expression (pCK53, $P_{CRE}$-SEAP-pA$_{bGH}$) (middle). Commercial GLP-1 (7-36) was used as a positive control. SEAP measurement was performed 24 h after supernatant transfer or commercial GLP-1 (7-36) treatment (right). Data are presented as means ± s.d. of n = 4 biologically independent samples, and p values were calculated using a two-tailed, unpaired Student's t-test. Source data are provided as a Source Data file.

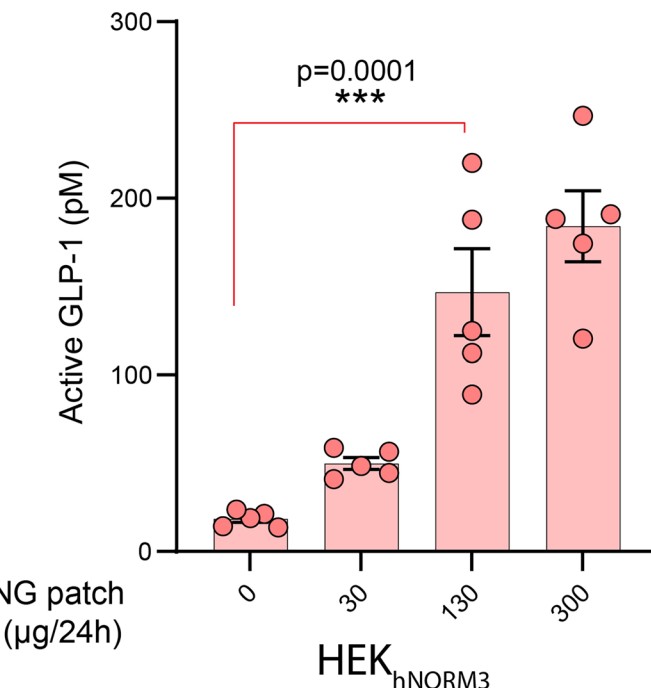

**Extended Data Fig. 5 | Serum levels of hNORM-regulated GLP-1 in response to different dosages of NG transdermal patches.** Alginate-PLL-alginate encapsulated HEK$_{hNORM3}$ cells (5x10$^6$ cells per mouse) were implanted subcutaneously into db/db mice (see methods). At 24 h after device implantation, different dosages of NG patches were topically applied above the cell implant. Active GLP-1 levels in the blood were measured 24 h following the drug administration. Data are presented as means ± SEM, n = 5, and p values were calculated using a two-tailed, unpaired Student's t-test. Source data are provided as a Source Data file.

a

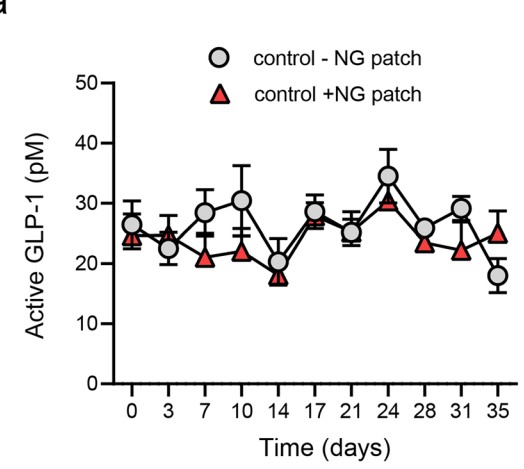

b

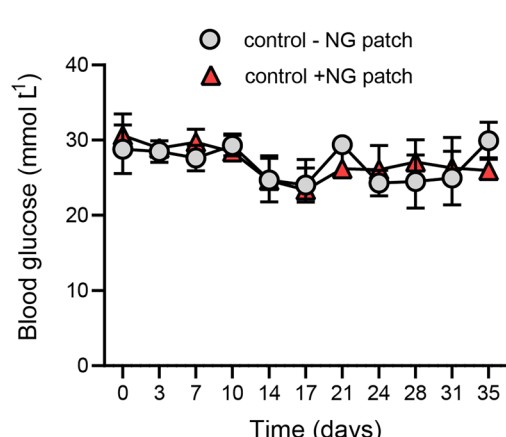

**Extended Data Fig. 6 | Plasma levels of GLP-1 and fasting glucose concentrations in control animals are not affected by NG transdermal patches.** Non-engineered WT HEK-293 cells were subcutaneously implanted in db/db mice as described in Fig. 4b, c. NG patches (130 µg/24 h) were topically applied just above the implant every two days for 35 days. Plasma GLP-1 (**a**) and fasting glucose (**b**) levels were analyzed every three days during the 35 days. Data are presented as means ± SEM of n = 4. Source data are provided as a Source Data file.

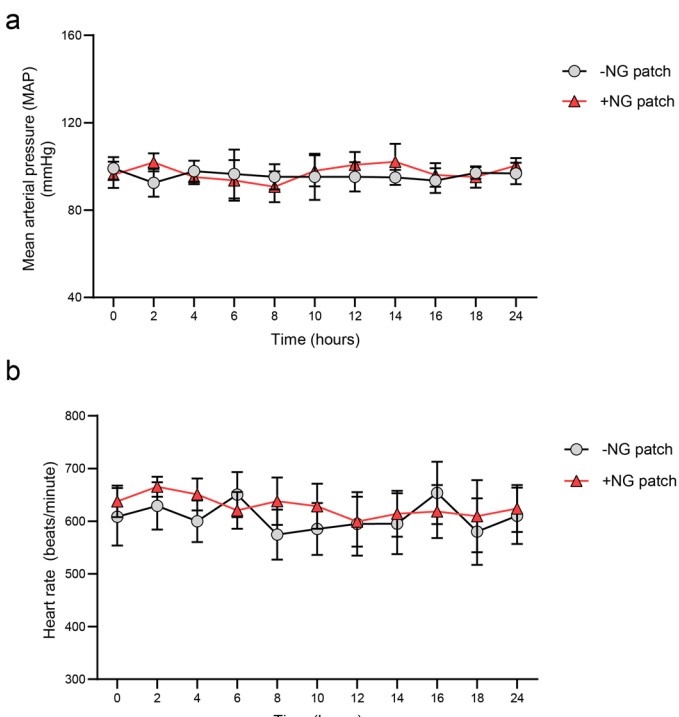

**Extended Data Fig. 7 | Continuous monitoring of blood pressure and heart rate of db/db mice following 35 days of hNORM-regulated GLP-1 treatment.** db/db mice were subcutaneously injected with alginate-PLL-alginate encapsulated HEK$_{hNORM3}$ cells (5x10$^6$ cells/mouse) as described in Fig. 5b. The treated groups received NG patches (130 μg/24 h) once every two days for 35 days. Following 35 days of treatment, continuous analysis of blood pressure (**a**), and heart rate (**b**) was performed for 24 h. Data are presented as mean ± SEM, n = 3. Statistical significance was analyzed by two-way ANOVA and p values were calculated using Tukey's multi-comparison tests. Source data are provided as a Source Data file.

|---|---|

# Reporting Summary

## Statistics

For all statistical analyses, confirm that the following items are present in the figure legend, table legend, main text, or Methods section.

| n/a | Confirmed | |
|---|---|---|
| ☐ | ☒ | The exact sample size (*n*) for each experimental group/condition, given as a discrete number and unit of measurement |
| ☐ | ☒ | A statement on whether measurements were taken from distinct samples or whether the same sample was measured repeatedly |
| ☐ | ☒ | The statistical test(s) used AND whether they are one- or two-sided<br>*Only common tests should be described solely by name; describe more complex techniques in the Methods section.* |
| ☒ | ☐ | A description of all covariates tested |
| ☐ | ☒ | A description of any assumptions or corrections, such as tests of normality and adjustment for multiple comparisons |
| ☐ | ☒ | A full description of the statistical parameters including central tendency (e.g. means) or other basic estimates (e.g. regression coefficient) AND variation (e.g. standard deviation) or associated estimates of uncertainty (e.g. confidence intervals) |
| ☐ | ☒ | For null hypothesis testing, the test statistic (e.g. *F*, *t*, *r*) with confidence intervals, effect sizes, degrees of freedom and *P* value noted<br>*Give P values as exact values whenever suitable.* |
| ☒ | ☐ | For Bayesian analysis, information on the choice of priors and Markov chain Monte Carlo settings |
| ☒ | ☐ | For hierarchical and complex designs, identification of the appropriate level for tests and full reporting of outcomes |
| ☒ | ☐ | Estimates of effect sizes (e.g. Cohen's *d*, Pearson's *r*), indicating how they were calculated |

*Our web collection on statistics for biologists contains articles on many of the points above.*

## Software and code

Policy information about availability of computer code

| Data collection | Absorbance, luminescence, and fluorescence data were collected using TECAN AG, Maennedorf, Switzerland. Western blot images were developed using FUSION Pulse TS (cat. no.37480003, Vilber, France). Blood glucose concentrations measured by a clinically licensed glucometer (Accu-Check Instant, Roche). Blood pressure and heart rate were analyzed using a computerized, non-invasive tail-cuff system (Bioseb, France, cat. no. BP-2000). RNA libraries were pooled and sequenced SR101 on an Illumina NovaSeq 6000 system. |
|---|---|
| Data analysis | GraphPad Prism 9 and adobe illustrator 2021. Statistical analysis of deep sequencing was performed with R scripts using edgeR ( v4.0.16). |

For manuscripts utilizing custom algorithms or software that are central to the research but not yet described in published literature, software must be made available to editors and reviewers. We strongly encourage code deposition in a community repository (e.g. GitHub). See the Nature Portfolio guidelines for submitting code & software for further information.

## Data

Policy information about [availability of data](availability of data)

All manuscripts must include a [data availability statement](data availability statement). This statement should provide the following information, where applicable:
- Accession codes, unique identifiers, or web links for publicly available datasets
- A description of any restrictions on data availability
- For clinical datasets or third party data, please ensure that the statement adheres to our [policy](policy)

> The main data supporting the results in this study are available within the paper and its Supplementary Information. Source data for the figures are provided with this paper. All raw and analysed data generated during the study are available from the corresponding author on reasonable request.

## Research involving human participants, their data, or biological material

Policy information about studies with [human participants or human data](human participants or human data). See also policy information about [sex, gender (identity/presentation), and sexual orientation](sex, gender (identity/presentation), and sexual orientation) and [race, ethnicity and racism](race, ethnicity and racism).

| | |
|---|---|
| Reporting on sex and gender | The study did not involve human research participants. |
| Reporting on race, ethnicity, or other socially relevant groupings | – |
| Population characteristics | – |
| Recruitment | – |
| Ethics oversight | – |

Note that full information on the approval of the study protocol must also be provided in the manuscript.

# Field-specific reporting

Please select the one below that is the best fit for your research. If you are not sure, read the appropriate sections before making your selection.

☒ Life sciences          ☐ Behavioural & social sciences          ☐ Ecological, evolutionary & environmental sciences

For a reference copy of the document with all sections, see [nature.com/documents/nr-reporting-summary-flat.pdf](nature.com/documents/nr-reporting-summary-flat.pdf)

# Life sciences study design

All studies must disclose on these points even when the disclosure is negative.

| | |
|---|---|
| Sample size | No statistical methods were used to predetermine sample size. However, based on our previous experiments and the existing literature with similar settings, when we used the same sample number of control and treated group and assuming type-I error = 0.05 and the probability of a type-II error = 0.20, we required at least 2.6 samples for each group to detect a 50% change between groups means. Therefore, we used at least 3 replicates for each group (Bai et al., Nat Med., 2019). |
| Data exclusions | No data were excluded. |
| Replication | All experiments were successfully repeated at least three times. |
| Randomization | Samples were randomly allocated into different experimental groups. For each mouse study, animals of the same genetic background were randomly allocated into different experimental groups. |
| Blinding | The investigators were not blinded to allocation during the experiments and outcome assessment. Blinding was not possible, as the same investigator processed the experiments and analysed the data. |

# Reporting for specific materials, systems and methods

We require information from authors about some types of materials, experimental systems and methods used in many studies. Here, indicate whether each material, system or method listed is relevant to your study. If you are not sure if a list item applies to your research, read the appropriate section before selecting a response.

## Materials & experimental systems

| n/a | Involved in the study |
|---|---|
| ☐ | ☒ Antibodies |
| ☐ | ☒ Eukaryotic cell lines |
| ☒ | ☐ Palaeontology and archaeology |
| ☐ | ☒ Animals and other organisms |
| ☒ | ☐ Clinical data |
| ☒ | ☐ Dual use research of concern |
| ☒ | ☐ Plants |

## Methods

| n/a | Involved in the study |
|---|---|
| ☒ | ☐ ChIP-seq |
| ☒ | ☐ Flow cytometry |
| ☒ | ☐ MRI-based neuroimaging |

## Antibodies

**Antibodies used**  The following primary antibodies were used in this study (all in 1:1000 dilution): monoclonal anti-ALDH2 antibody produced in rabbit (Cell Signaling, cat. no. 18818) and mouse anti-α-actinin (Cell Signaling, cat. no. 69758, clone E7U1O). Secondary HRP-conjugated goat anti-rabbit (cat. no. 111-035-144, polyclonal) and anti-mouse (cat. no. 115-035-003, polyclonal) antibodies were purchased from Jackson Immunoresearch, West Grove, PA, and used at a dilution of 1:10,000 dilution.
For ICH, we used leaved Caspase 3 antibody (Cell Signaling Technology, Lot Nr.: 47, Catalog Nr.: 9661S, in 1:500 dilution)

**Validation**  Anti-ALDH2 antibody was validated by transfection of HEK-293 cells with human ALHD2 encoding expression vector followed by immunoblotting.
Mouse anti-α-actinin was validated by loading different amounts of cell lysate. A corelation between the lysate amount and band intensity in the correct protein size was noticed.
Both secondary antibodies were validated by the absence of bands/signal in immunoblot that has not been exposed to primary antibody. Cleaved Caspase-3 antibody was validated by specific staining of a control human tonsil and rat spleen. The technical negative controls showed no staining in the capsules that could be attributed to unspecific binding of the used antibody.

## Eukaryotic cell lines

Policy information about cell lines and Sex and Gender in Research

**Cell line source(s)**  Human embryonic kidney cells (HEK-293T, ATCC: CRL-11268), HeLa cells (HeLa, ATCC: CCL-2), HepG2 cells (HEPG2, ATCC: HB-8065), baby hamster kidney cells (BHK, ATCC: CCL-10), and HT-1080 cells (HT1080, ATCC: CCL-121), Chinese hamster ovary cells (CHOK1, ATCC: CCL-61), A549 cells (A549, ATCC: CCL-185), and Primary dermal fibroblast normal; human, neonatal (HDFn; ATCC, PCS-201-010).

**Authentication**  Cells were authenticated by ATCC. All the phenotypes of cell lines were frequently checked and controlled by microscopy.

**Mycoplasma contamination**  Cells were tested frequently for mycoplasma, and confirmed as negative.

**Commonly misidentified lines** (See ICLAC register)  No commonly misidentified cell lines were used.

## Animals and other research organisms

Policy information about studies involving animals; ARRIVE guidelines recommended for reporting animal research, and Sex and Gender in Research

**Laboratory animals**  12-week-old male C57BL/6 or db/db mice (BKS.Cg-Dock7m +/+ Leprdb/J, derived from C57BL/6J, Janvier Labs Saint-Berthevin, France) were used.

**Wild animals**  The study did not involve wild animals.

**Reporting on sex**  Male mice were used, due to husbandry convenience, as supported by: Bai et al., Nature Medicine,2019; Krawczyk et al., Science, 2020; Chen et al, Nature Chemical Biology; and Zhou et al., Nature Biotechnology, 2021. Sex was not considered in the study design. No data disaggregated for sex were collected.

**Field-collected samples**  The study did not involve samples collected from the field.

**Ethics oversight**  All animal experiments were conducted in compliance with the directive of the European Community Council (2010/63/EU) and approved by the French Republic (project no. DR2013-v2, licenses: Ghislaine Charpin-El Hamri, no. 69266309, and Shuai Xue, no. LTK4899). In addition, experiments were also performed following the Animal Care Guidelines of the Ministry of Science and Technology of the People's Republic of China and were approved by the Institutional Animal Care and Use Committee (IACUC) of Westlake University (Protocol ID: AP#24-088-XMQ). We adhered to rigorous ethical standards to ensure the welfare of the animals involved, and our study design was guided by the principles of the 3Rs: replacement, reduction and refinement.

Note that full information on the approval of the study protocol must also be provided in the manuscript.

