## [Peer Review File · Nature Biomedical Engineering]

Nitroglycerin-responsive gene switch for the on-demand production of therapeutic proteins

Corresponding author: Martin Fussenegger

Editorial note

This document includes relevant written communications between the manuscript's corresponding author and the editor and reviewers of the manuscript during peer review. It includes decision letters relaying any editorial points and peer-review reports, and the authors' replies to these (under 'Rebuttal' headings). The editorial decisions are signed by the manuscript's handling editor, yet the editorial team and ultimately the journal's Chief Editor share responsibility for all decisions.

Any relevant documents attached to the decision letters are referred to as **Appendix #**, and can be found appended to this document. Any information deemed confidential has been redacted or removed. Earlier versions of the manuscript are not published, yet the originally submitted version may be available as a preprint. Because of editorial edits and changes during peer review, the published title of the paper and the title mentioned in below correspondence may differ.

Correspondence

Thu 07 Dec 2023

Decision on Article nBME-23-2496

Dear Prof Fussenegger,

Thank you again for submitting to *Nature Biomedical Engineering* your manuscript, "Treatment of diabetes and obesity by a human nitric oxide-regulated gene switch percutaneously controlled by nitroglycerin patches". The manuscript has been seen by 4 experts, whose reports you will find at the end of this message.

You will see that the reviewers appreciate aspects of the work. However, they articulate concerns about the degree of support for some of the claims and about the advance that the work represents over relevant published studies, and provide useful suggestions for improvement. We hope that with significant further effort you can address the criticisms, increase the level of significance of the study, and convince the reviewers of its merits. In particular, we would expect that a revised version of the manuscript provides:

- * Longer-term characterization of the functionality and safety (including any local and systemic immune responses) of the microencapsulated designer cells, as per the comments from Reviewers #2 and #3.
- * Characterization of the expression levels of the produced proteins, and of the kinetics of the system's response to the presence of nitric oxide.
- * Discussion of other relevant application possibilities, and inclusion of caveats as to the foreseeable translatability shortcomings of this strategy for the production of GLP-1.

When you are ready to resubmit your manuscript, please upload the revised files, a point-by-point rebuttal to the comments from all reviewers, the reporting summary, and a cover letter that explains the main improvements included in the revision and responds to any points highlighted in this decision.Please follow the following recommendations:

- * Clearly highlight any amendments to the text and figures to help the reviewers and editors find and understand the changes (yet keep in mind that excessive marking can hinder readability).
- * If you and your co-authors disagree with a criticism, provide the arguments to the reviewer (optionally, indicate the relevant points in the cover letter).
- * If a criticism or suggestion is not addressed, please indicate so in the rebuttal to the reviewer comments and explain the reason(s).
- * Consider including responses to any criticisms raised by more than one reviewer at the beginning of the rebuttal, in a section addressed to all reviewers.
- * The rebuttal should include the reviewer comments in point-by-point format (please note that we provide all reviewers with the reports as they appear at the end of this message).
- * Provide the rebuttal to the reviewer comments and the cover letter as separate files.

We hope that you will be able to resubmit the manuscript within 25 weeks from the receipt of this message. If this is the case, you will be protected against potential scooping. Otherwise, we will be happy to consider a revised manuscript as long as the significance of the work is not compromised by work published elsewhere or accepted for publication at *Nature Biomedical Engineering*.

We hope that you will find the referee reports helpful when revising the work. Please do not hesitate to contact me should you have any questions.

Best wishes,

Filipe

Dr Filipe Almeida
Associate Editor, Nature Biomedical Engineering

Reviewer #1 (Report for the authors (Required)):

Mahameed et al. describe the design of a nitric oxide-inducible gene expression system. In brief, this fully human gene switch consists of human cALHD2 for the local conversion of NG into the trigger compound NO, the sGC for the production of the second messenger cyclic GMP (cGMP), and human cGMP-dependent PKG1, which triggers a cGMP signal transduction cascade, activating gene expression driven by a CREB1-cognate synthetic promoter. The system was fully human and functionally characterized, including dynamic ranges and real-time systemic dosing. Furthermore, it was customized for the expression of the therapeutic protein GLP-1 to treat diabetes by topically percutaneous application of clinically licensed NG patches above the subcutaneously microencapsulated hNORM transgenic human cell implant site.

In general, this is an interesting work in the field of controllable cell-based precision therapy. While the system appears to be functional and thus has clear applications in cell therapy, there are several major limitations of the system that the authors need to address before publication.

1. The authors employ a patch for transdermal delivery of trigger compounds, inducing the implanted cells to produce GLP-1 and reduce blood glucose levels. Notably, a glucose-responsive insulin patch designed to regulate blood glucose has been published in *Nature Biomedical Engineering* (2020, PMID 32015407). The authors should compare their subcutaneous transplantation method for treating diabetes with the direct delivery of insulin via an insulin patch to determine which approach is more convenient. Additionally, they should consider selecting a more appropriate disease scenario to showcase the advantageous aspects of

their cell therapy method.

2. The system's dependence on endogenous central signaling pathways makes it non-orthogonal to the mammalian cells used, potentially leading to adverse effects related to cGMP, PKG, and other factors. To evaluate the impact on target cells and assess the significance of the research findings regarding its potential applications in cell therapy, conducting a gene expression study (RNASeq) is crucial.
3. Nevertheless, the short effective distance between the NG patches and the implanted cells poses a challenge for the system's applicability in deep tissue applications. Moreover, hNORM shows no regulatory efficacy when administered through intraperitoneal injection of NG, further limiting its utility. It is worth considering the performance of this system in larger animal models, such as rats, to better understand its potential in an in vivo setting.
4. The authors used a method involving encapsulation of HEK-293 tumor cell lines in microcapsules and tested their effects in mice for one month. I wonder, after one month, if the cells inside the microcapsules are still alive despite being unable to proliferate and undergoing apoptosis. The authors should assess the viability of the cells transplanted in vivo for one month. Additionally, does the transplantation of microcapsules lead to fibrosis issues, and can the cells still secrete the expressed proteins?
5. In the treatment of metabolic diseases, particularly diabetes, continuous monitoring of blood glucose levels over an extended period is crucial. For example, including a 24-hour continuous blood glucose data in this study would strengthen its significance. This recommendation also extends to safety assessment, which should incorporate measurements of heart rate and blood pressure. Furthermore, the authors have not provided information concerning the local concentration of NO and its sustained metabolic levels in vivo. Including these details would greatly enhance readers' understanding and the practical applicability of the findings.
6. The authors did not perform a dose exploration in their animal experiments; instead, they directly selected a dosage of 130 $\mu\text{g}/24\text{h}$. It would be helpful to understand the rationale behind this choice and whether any attempts were made to explore lower or higher dosages. Additionally, if a different animal model were to be used, how would the appropriate dosage be determined?
7. The authors have commendably acknowledged the importance of considering the immunogenicity of this technology for in vivo applications. Therefore, it is crucial to explore the use of safer cell types in future applications. I recommend that the authors test the functionality of this system in a broader range of non-cancerous cells to showcase its applicability across a wider spectrum.

Minor points:

1. As stated in line 241 "Since NO at high local concentrations can introduce post-translational modifications such as nitrosylation and tyrosine nitration, which may alter protein activity." Based on this information, we suspect that the dosage of NG patches (130 $\mu\text{g}/24\text{h}$) may be toxic, and additional data is needed to address this concern. Moreover, considering that the concentration of NG patches used (130 $\mu\text{g}/24\text{h}$) is relatively high, it would be beneficial to explore approaches to enhance system sensitivity.
2. The conversion of NO concentration by NG patches, facilitated by endogenous aldehyde dehydrogenase 2 (ALDH2), should be measured, and the dosage of NO delivered to the topical skin needs to be accurately calculated. It is important to ensure that the calculated dosage is safe and non-toxic.
3. Fig. 3e requires the addition of complete statistical information, and Fig. 6d should include the corresponding insulin curve, which can be placed in the supplementary materials.
4. The sentence in line 255 "the control mice with parent cell implant" is not clear.
5. Page 6, the subtitles in line 175 and line 189 are the same. This is misleading.
6. The figure caption in Fig. 2c "DETA NONOate" should maintain consistency.
7. Please clarify specific advantages of hNORM in line 353.

8. The authors should provide the related amino acid or DNA sequences of hNORM.

Reviewer #2 (Report for the authors (Required)):

The paper by Mahameed et al. presents the design and optimization of a human nitric oxide-responsive transgene regulation modality (hNORM) in mammalian cells. The researchers successfully established a stable hNORM-transgenic human cell line (HEKhNORM1) and later improved its performance by engineering another cell line (HEKhNORM2) for additional ALDH2 expression. Utilizing topical nitroglycerin patches to control NO production, authors demonstrated precise control of transgene expression in subcutaneously implanted microencapsulated cells in mice. In a proof-of-concept experiment, they showed that hNORM-controlled GLP-1 expression restores blood-glucose homeostasis in a mouse model of type-2 diabetes without causing cardiovascular interference.

While the conceptual framework is intriguing, several critical aspects necessitate further elucidation.

Degree of Advance:

The work presents a technological and methodological advancement in gene regulation for therapeutic purposes. However, the translational readiness of this system, particularly concerning long-term safety, drug interactions, and scalability, appears lacking when compared to the existing state of the art in the field.

1. The manuscript lacks quantitative PCR (qPCR) or RNA sequencing data to confirm the expression levels of sGC, PKG1, NLuc, and ALDH2.
2. Although cell viability is mentioned, explicit cytotoxicity assays under various conditions are missing.
3. A comparative evaluation of SEAP and NLuc to validate the switch in sensitivity and specificity is lacking. This analysis could provide a stronger rationale for the choice of NLuc.
4. The manuscript displays good reversibility to NO donors but does not elucidate the precise response time for system activation and deactivation. Detailed kinetics concerning the hNORM system's response to NO are important for assessing its clinical utility.
5. While the authors claim the cell line is stable, the manuscript lacks data on long-term stability across multiple passages.
6. Given ALDH2's role in drug metabolism, how would the concurrent administration of drugs that inhibit or induce ALDH2 affect your system?
7. The incorporation of additional enzymes augments the system's complexity which could impede its use and optimization.
8. The paper lacks discussion on how this system could be scaled for industrial or clinical applications.
9. The in vivo study covers 35 days, but long-term effects of the system remain unknown. Do the cells maintain their functionality over extended periods?
10. The study does not specify the stability or longevity of the encapsulation materials (alginate-PLL-alginate capsules). Can these capsules maintain their structural integrity and functionality over a long period?
11. There is no mention of whether these capsules elicit a foreign body reaction, which is a key concern in any implantable device. Any reaction could affect the function of the encapsulated cells and therefore the efficacy of the system.
12. Since many type-2 diabetes patients are on multiple medications, it would be useful to study how your system interacts with these commonly used drugs. This is particularly important given the central role of ALDH2 in drug metabolism.

Minor Comments:

1. No histological analyses of the explanted encapsulated cells are provided. This could show how the system interacts with surrounding tissues and whether it induces any adverse reactions or fibrosis.
2. While the discussion talks about the advantages and the broader context, it doesn't adequately discuss the limitations of the study itself. For example, the dependency on ALDH2 and its implications are not discussed here.
3. A more detailed comparison to existing therapies, especially in terms of efficacy, safety, and cost-effectiveness, would strengthen the discussion.
4. While the paper talks about how the system could be used in chronic conditions, it does not discuss how it compares to existing treatments for those conditions in terms of effectiveness, safety, and cost.

Reviewer #3 (Report for the authors (Required)):

In this study, the authors have developed a new method, referred to as human Nitric Oxide-Responsive Transgene Regulation Modality (hNORM), which allows for the controlled production and distribution of therapeutic proteins in response to NO. This response is triggered by the enzymatic conversion of nitroglycerin (NG), delivered percutaneously through clinically approved patches. To demonstrate its feasibility, they applied this system to deliver GLP-1 through subcutaneously implanted microencapsulated hNORM-transgenic GLP-1-expressing human cells controlled by licensed dermal patches placed above. This device offers intriguing potential for the modulation of targeted protein expression via an NG patch. Nevertheless, several concerns must be addressed before potential acceptance.

1. The system exhibits a protracted response time, constraining its utility. Would the authors consider proposing a more viable application scenario beyond GLP-1 deployment?
2. The survival metrics of HEK-hNORM cells post-implantation remain unquantified. It is imperative to determine whether these cells succumb to apoptosis or other cell death over extended periods post-implantation and to assess the subsequent impact on the immune microenvironment at the implantation site and the broader immune system response.
3. The selection of HEK-293 cells for in vivo implantation raises concerns. The authors are urged to conduct comprehensive analyses to confirm the absence of systemic toxicity and to rule out local immune cell infiltration at the implantation site in the treated animals. Furthermore, it is crucial to investigate whether there are discernible differences in the NO response in cells pre- and post-implantation.

Minor issues:

4. The manuscript exhibits several formatting inconsistencies. Specifically, the indentation of paragraphs, such as those beginning on lines 154 and 177, is irregular; line 189 contains a superfluous subheading; the abbreviation "Fig" in line 258 is not appropriately bolded; and the citation format for articles, as seen in references 6, 25, 31, and 43, deviates from the standard.
5. The captions of figures do not conform to accepted standards and necessitate revision, such as "Figure 3 | hNORM is effective, tunable, and reversible, and is blocked by methylene blue."

Reviewer #4 (Report for the authors (Required)):

The authors describe a nitric oxide(NO)-responsive transgene regulation modality (hNORM) to control the dynamic production and delivery of protein therapeutics. The strategy is based on engineered enzymatic reaction platform converting nitroglycerin (NG), that is transdermally delivered by clinically licensed patches, to NO. The authors have previously reported a series of in situ transgene expression & switching systems, based upon synthetic biological engineering approach, that are potentially relevant to the future gene- and cell-based therapies. In the present report, the essence lies on the choice of NG, which has been widely used to control heart conditions, in combination with a clinically licensed NG patches. To me, this is a wise and convincing combination in the context of clinical translation. Overall, the manuscript is written well

including the concept, design, experimental detail and results and properly discussed. I did not find any particular technical issues in the manuscript. Therefore, I recommend the paper be considered publication in Nature Biomedical Engineering after some improvements.

I have two suggestions to strengthen the paper:

1. There is no real image shown at all, which makes it a bit less eye-appealing. Please consider to provide some for device specification, implantation site, in vivo or ex vivo NG biodistribution imaging, and etc.
2. In discussion please provide any limitations & issues to be addressed before clinical translation.

Minor issue:

- I. 143: Should “PGK2-expressing” be “PKG2-expressing”?

Mon 05 Aug 2024

Decision on Article NBME-23-2496A

Dear Professor Fussenegger,

Thank you for your revised manuscript, "Treatment of diabetes and obesity by a human nitric oxide-regulated gene switch percutaneously controlled by nitroglycerin patches". Having consulted with Reviewers #1, #2 and #3 - unfortunately reviewer #4 wasn't able to deliver a report, and is by now is unlikely that will, I am pleased to write that we shall be happy to publish the manuscript in *Nature Biomedical Engineering*.

We will be performing detailed checks on your manuscript, and in due course will send you a checklist detailing our editorial and formatting requirements. You will need to follow these instructions before you upload the final manuscript files.

Best wishes,
Filipe

Dr Filipe Almeida
Senior Editor, Nature Biomedical Engineering

Reviewer #1 (Report for the authors (Required)):

They have addressed most of my concerns.

In the title and the context, instead of using the term "treatment" for diabetes or obesity, they should say "controlling blood glucose levels" and "managing or controlling body weight." This method is still far from treating diabetes and obesity.

Reviewer #2 (Report for the authors (Required)):

The revised manuscript by Mahameed et al. has substantially improved, addressing key concerns with comprehensive new data and analyses. I recommend this manuscript for publication as it stands, believing it will contribute valuable knowledge to the field.

Reviewer #3 (Report for the authors (Required)):

The authors have well addressed my concerns.

Rebuttal 1

nBME-23-2496.R1 - Point-by-point responses

Reviewer 1

Mahameed et al. describe the design of a nitric oxide-inducible gene expression system. In brief, this fully human gene switch consists of human cALHD2 for the local conversion of NG into the trigger compound NO, the sGC for the production of the second messenger cyclic GMP (cGMP), and human cGMP-dependent PKG1, which triggers a cGMP signal transduction cascade, activating gene expression driven by a CREB1-cognate synthetic promoter. The system was fully human and functionally characterized, including dynamic ranges and real-time systemic dosing. Furthermore, it was customized for the expression of the therapeutic protein GLP-1 to treat diabetes by topically percutaneous application of clinically licensed NG patches above the subcutaneously microencapsulated hNORM transgenic human cell implant site.

In general, this is an interesting work in the field of controllable cell-based precision therapy. While the system appears to be functional and thus has clear applications in cell therapy, there are several major limitations of the system that the authors need to address before publication.

Thank you for your constructive feedback and suggestions for enhancing the quality of our work.

1. The authors employ a patch for transdermal delivery of trigger compounds, inducing the implanted cells to produce GLP-1 and reduce blood glucose levels. Notably, a glucose-responsive insulin patch designed to regulate blood glucose has been published in *Nature Biomedical Engineering* (2020, PMID 32015407). The authors should compare their subcutaneous transplantation method for treating diabetes with the direct delivery of insulin via an insulin patch to determine which approach is more convenient. Additionally, they should consider selecting a more appropriate disease scenario to showcase the advantageous aspects of their cell therapy method.

We regret our oversight in not citing the work by Yu et al. (2020 *Nature Biomedical Engineering* 4:499). In the revised manuscript, we cited and highlighted the impact and potential of this elegant work within the discussion section.

However, it should be noted that Yu et al. employed a pre-loaded polymer-based microneedle patch for glucose-sensitive real-time in-situ percutaneous delivery of **insulin** in the context of closed-loop treatment for **type-1 diabetes**. In contrast, our approach involves a genetically encoded cell-based therapy utilizing a clinically licensed patch for transdermal delivery of nitroglycerin. This delivery triggers the real-time production of biopharmaceuticals, exemplified by **GLP-1**, for open-loop treatment of experimental **type-2 diabetes and obesity**.

Given the fundamentally different nature and targets of our methodologies, we think a direct comparison is not appropriate. However, we believe the complementary nature of these different strategies will advance percutaneous control modalities for biopharmaceuticals, and both should be of interest to readers of *Nature Biomedical Engineering*.

We wish to emphasize that our selection of GLP-1 for validation of our novel nitroglycerin/nitric-oxide-tunable gene switch is strategic. GLP-1 and its derivatives, including the blockbuster drugs Ozempic, Saxenda, and Wegovy, are pivotal in managing type-2 diabetes and obesity. These conditions have a global impact, affecting over 10% of the population today, as reported by the International Diabetes Federation (diabetesatlas.org). Moreover, their prevalence is expected to rise, with nearly one-quarter of the human population projected to be affected by 2035, according to the World Obesity Atlas 2023 (www.worldobesity.org).

We incorporate these points in the discussion section of our revised manuscript.

2. The system's dependence on endogenous central signaling pathways makes it non-orthogonal to

the mammalian cells used, potentially leading to adverse effects related to cGMP, PKG, and other factors. To evaluate the impact on target cells and assess the significance of the research findings regarding its potential applications in cell therapy, conducting a gene expression study (RNASeq) is crucial.

The choice between employing an orthogonal gene switch and utilizing endogenous cell signaling is a perennial debate. While orthogonal switches are less susceptible to signaling interference, they often necessitate the use of molecular components from other species, raising safety and immunogenicity concerns. Conversely, endogenous gene-control modalities, despite being prone to interference to varying degrees, are generally considered safer and more biocompatible, making them often a preferred option for therapeutic purposes.

An illustrative example is the well-known CAR-T therapy, where engineered CAR receptors are embedded in the cell's endogenous signaling. Despite the potential for signaling interference, CAR-T therapy has obtained clinical licensing for hematopoietic malignancies, highlighting the acceptance of such approaches in therapeutic contexts (Cappell et al., 2023, *Nature Reviews Clinical Oncology* 20:359). Furthermore, we and others have reported several *in vivo* studies for the treatment of prominent experimental diseases, capitalizing on endogenous control components. Non-limiting examples include: Miller et al., 2021, *Nature Biomedical Engineering* 5:1348; Bai et al., 2019, *Nature Medicine* 25:1266; Huang et al., 2023 *Nature Metabolism* 5: 1395; Kemmer et al., 2010, *Nature Biotechnology* 28:355; Liu et al., 2018, *Cell* 174:259; Krawczyk et al., 2020, *Science* 368:993; Stanley et al., 2012, *Science* 336:604; Stanley et al., 2015, *Nature Medicine* 21:92; Stanley et al., 2016, *Nature* 531:647; Wang et al., 2018, *Nature Biomedical Engineering* 2:114; Xie et al., 2016, *Science* 354:1296; Ye et al., 2017, *Nature Biomedical Engineering* 1:5; Ye et al., 2011, *Science* 332:1565; Zhao et al., 2023, *Lancet Diabetes and Endocrinology* 11:637; Zhou et al., 2022, *Nature Biotechnology* 40:262. For an overview of this topic, see Xie and Fussenegger, 2018; *Nature Reviews Molecular Cell Biology* 19:507.

Moreover, our data in **Fig. 6e** and **6f** demonstrate that unstimulated hNORM3 performance is comparable to controls, suggesting minimal interference *in vivo*. This observation supports the feasibility of utilizing endogenous gene circuits, indicating their potential for safe and effective use in therapeutic applications.

Nevertheless, to explore potential changes in gene expression induced by the operation of our nitric oxide gene switch, we conducted the requested gene expression (RNASeq) study at the commencement and conclusion of the 35-day *in vivo* experiment shown in **Fig. 5b** and **5c**. This encompasses mRNA sequencing for both the implanted designer cells and the tissue surrounding the implant site as depicted in **Fig. S12** (please also refer to our response to point 1 of reviewer 2). In this analysis, we found no significant difference in the P-values of PCA analysis between the treated and the untreated groups, which implies a low or negligible level of interference.

3. Nevertheless, the short effective distance between the NG patches and the implanted cells poses a challenge for the system's applicability in deep tissue applications. Moreover, hNORM shows no regulatory efficacy when administered through intraperitoneal injection of NG, further limiting its utility. It is worth considering the performance of this system in larger animal models, such as rats, to better understand its potential in an *in vivo* setting.

In practical use, the short effective reach of percutaneous control in systemic biopharmaceutical delivery through a dermal patch represents a notable advantage—see for example Yu et al., 2020 *Nature Biomedical Engineering* 4:499. It contributes to improved patient convenience and compliance, minimizes the risk of systemic side effects and interference, and enhances bioavailability (please also refer to our response to point 2 above).

The significance of the control experiment depicted in **Fig. 4b** lies in its demonstration that hNORM activation, induced by short-lived nitric oxide, is confined to the subcutaneous space beneath the dermal nitroglycerin patch. This crucial experiment establishes that dermal nitroglycerin patches

neither influence distant endogenous physiological activities nor impede endogenous nitric oxide signaling that may affect GLP-1 expression by the implanted cells. This validation provides strong support for our rationale for employing non-orthogonal endogenous control components, as detailed in our response to comment 2 by this reviewer.

Moreover, extensive research by us and others has documented challenges associated with systemic deep-tissue control modalities, in contrast to percutaneous control. These challenges have been described in many publications, including but not limited to the following: Miller et al., 2021, *Nature Biomedical Engineering* 5:1348; Bai et al., 2019, *Nature Medicine* 25:1266; Huang et al., 2023 *Nature Metabolism* 5: 1395; Kemmer et al., 2010, *Nature Biotechnology* 28:355; Liu et al., 2018, *Cell* 174:259; Krawczyk et al., 2020, *Science* 368:993; Stanley et al., 2012, *Science* 336:604; Stanley et al., 2015, *Nature Medicine* 21:92; Stanley et al., 2016, *Nature* 531:647; Wang et al., 2018, *Nature Biomedical Engineering* 2:114; Xie et al., 2016, *Science* 354:1296; Ye et al., 2017, *Nature Biomedical Engineering* 1:5; Ye et al., 2011, *Science* 332:1565; Zhao et al., 2023, *Lancet Diabetes and Endocrinology* 11:637; Zhou et al., 2022, *Nature Biotechnology* 40:262. We clarify these points in the revised version of our manuscript.

Regarding the suggestion to replicate our experiments in a larger animal, we feel that such an experiment would be difficult to justify in the light of current European legislation (Directive 2010/63/EC), which urges researchers to adopt approaches that minimize the use of animals for a given research objective, emphasizing the principles of replacement, reduction, and refinement (3R) of animal experiments throughout the European Union and Switzerland. In this context, the db/db mouse model is a widely accepted model for treatment studies in experimental type-2 diabetes and obesity, and has been extensively utilized in both our work and that of others. Illustrative examples include Ye et al., 2011, *Science* 332:1565; Quarta et al., 2022, *Nature Metabolism* 4:1071; Gilroy et al., 2020, *Science Advances* 26:6; Ye et al., 2013, *PNAS* 110:18150; Xie et al., 2016, *Science* 354:1296; Xue et al., 2017, *Mol. Ther.* 25:443; Shao et al., 2017, *Science Transl. Med.* 26:9(387):eaal2298; Bojar et al., 2018, *Nature Communications* 9:2318; Mansouri et al., 2021, *Nature Communications* 12:3388, and Guha Ray et al., 2023, *Nature Communications* 14:3151. Therefore, we consider that the use of the db/db mouse model aligns not only with the scientific consensus, but also with current ethical and legal considerations. Thus, instead of introducing a larger animal model, we refined our current animal study to address reviewers' comments and suggestions.

4. The authors used a method involving encapsulation of HEK-293 tumor cell lines in microcapsules and tested their effects in mice for one month. I wonder, after one month, if the cells inside the microcapsules are still alive despite being unable to proliferate and undergoing apoptosis. The authors should assess the viability of the cells transplanted in vivo for one month. Additionally, does the transplantation of microcapsules lead to fibrosis issues, and can the cells still secrete the expressed proteins?

Alginate-microencapsulated transgenic HEK-293 cells have been widely employed to validate groundbreaking proof-of-concept ideas in mouse models of a variety of human diseases. Notably, alginate has provided positive outcomes in human clinical trials (Jacobs-Tulleneers-Thevissen et al., 2013, *Diabetologia* 56:1601; ClinicalTrials.gov NCT013797299), and it has been demonstrated that alginate-encapsulated mammalian cells exhibit prolonged survival and functionality in vitro, sometimes extending beyond a year. We recently reported a 5-week study (Krawczyk et al., 2020, *Science* 368:993; Bai et al., 2019; *Nature Medicine* 25:1266) of long-term viability, focusing on fibrosis associated with alginate-encapsulated HEK-293 cells in mice.

Despite the inherent association of fibrosis with implantation, recent advancements in modified alginate have made it possible to substantially delay its development (Bose et al., 2020, *Nature Biomedical Engineering* 4:814; Doloff et al., 2017, *Nature Materials* 16:671; Farah et al., 2019, *Nature Materials* 18:892). Crucially, as illustrated in **Fig. 5b**, we found that alginate-encapsulated hNORM-transgenic HEK-293 cells exhibited responsiveness to nitric oxide and consistently released therapeutic levels of GLP-1 throughout the entire 35-day study period.

We have included a comprehensive discussion of these issues, citing studies that validate the viability and functionality of alginate-encapsulated cells and address issues related to fibrosis in cell implants. To complement the RNAseq analysis (please refer to our response to point 2 from this reviewer) and immunological analyses (please see our responses to points 2 and 3 from reviewer 3), we conducted further assessments of cell viability and implant fibrosis (according to Krawczyk et al., 2020, *Science* 368:993; Bai et al., 2019, *Nature Medicine* 25:1266), as well as immuno-histochemical analyses (according to Huang et al., 2023 *Nature Metabolism* 5: 1395; Krawczyk et al., 2020, *Science* 368:993; Maity et al., 2023, *Advanced Materials* 35:e2300890) at both the initiation and conclusion of the 35-day experimental study. These histological data are presented in supplemental **Fig. S11** and these results show (i) insignificant local immune response, (ii) absence of fibrosis, (iii) negligible apoptosis of the encapsulated cells (as shown by anti-caspase-3 immunohistochemistry), and (iv) high integrity and stability of the capsules.

5. In the treatment of metabolic diseases, particularly diabetes, continuous monitoring of blood glucose levels over an extended period is crucial. For example, including a 24-hour continuous blood glucose data in this study would strengthen its significance. This recommendation also extends to safety assessment, which should incorporate measurements of heart rate and blood pressure. Furthermore, the authors have not provided information concerning the local concentration of NO and its sustained metabolic levels in vivo. Including these details would greatly enhance readers' understanding and the practical applicability of the findings.

In the revised version of the study, we conducted continuous 24-hour monitoring of blood glucose levels (new **Fig. 6g**), recorded blood pressure readings (new **Fig. S7, a**), and heart rate (new **Fig. S7, b**) as requested.

Furthermore, we quantified the levels of local nitric oxide release from NG transdermal patches in mice tissue at different intervals (according to Kulshrestha et al., 2019, *Biomed Pharmacother* 112:108571; Barmak et al., 2018, *J Med Life* 11:225, and Salama et al., 2007, *Liver Int.* 10.1111). In this experiment, we also assessed the contribution of ALHD2 in NG biotransformation to nitric oxide. These results are shown in new **Fig. 4e** (please also refer to point minor-2 of the same reviewer).

6. The authors did not perform a dose exploration in their animal experiments; instead, they directly selected a dosage of 130 $\mu\text{g}/24\text{h}$. It would be helpful to understand the rationale behind this choice and whether any attempts were made to explore lower or higher dosages. Additionally, if a different animal model were to be used, how would the appropriate dosage be determined?

The reviewer may have overlooked our dose explorations as illustrated in **Fig. 4b**, where we present nitroglycerin-dependent transgene expression within and below the therapeutically relevant nitroglycerin concentration range. This served to provide an experimental basis for the in vivo dosage of 130 $\mu\text{g}/24\text{h}$. The same approach should be employed to ascertain the appropriate concentration for any other animal model.

To ensure the dose-responsiveness of hNORM-engineered cells regardless of the transgene expressed, we replicated the experiment using the hNORM-transgenic GLP-1-expressing HEK-293 cells (HEK_{hNORM3} cells) in the animal study. The findings from this additional experiment are incorporated into a supplemental **Fig. S5**.

7. The authors have commendably acknowledged the importance of considering the immunogenicity of this technology for in vivo applications. Therefore, it is crucial to explore the use of safer cell types in future applications. I recommend that the authors test the functionality of this system in a broader range of non-cancerous cells to showcase its applicability across a wider spectrum.

We concur. We assessed the performance of hNORM in various cell types, including human primary dermal fibroblasts (HDFn), for which new data are added to **Fig. 2c** in the revised version.

Minor points:

1. As stated in line 241 “Since NO at high local concentrations can introduce post-translational modifications such as nitrosylation and tyrosine nitration, which may alter protein activity.” Based on this information, we suspect that the dosage of NG patches (130 µg/24 h) may be toxic, and additional data is needed to address this concern. Moreover, considering that the concentration of NG patches used (130 µg/24 h) is relatively high, it would be beneficial to explore approaches to enhance system sensitivity.

As noted in our response to point 4, we carried out a thorough analysis encompassing viability, implant fibrosis, immuno-histochemical assessment, cytokines profiling, and RNAseq at both the commencement and conclusion of the 35-day experimental study. In these experiments, we found no marked toxicity (**Fig. S10, Fig. S11, Fig. S12**).

Regarding system sensitivity, we have already enhanced the sensitivity of hNORM by more than three-fold through the overexpression of aldehyde dehydrogenase 2 (ALDH2), the enzyme responsible for converting nitroglycerin to nitric oxide, as illustrated in **Fig. 4b-d**.

2. The conversion of NO concentration by NG patches, facilitated by endogenous aldehyde dehydrogenase 2 (ALDH2), should be measured, and the dosage of NO delivered to the topical skin needs to be accurately calculated. It is important to ensure that the calculated dosage is safe and non-toxic.

We quantified the levels of local nitric oxide release from NG transdermal patches in mice tissue at different intervals (according to Kulshrestha et al., 2019, *Biomed Pharmacother* 112:108571; Barmak et al., 2018, *J Med Life* 11:225, and Salama et al., 2007, *Liver Int.* 10.1111). In this experiment, we also assessed the contribution of ALDH2 in NG biotransformation to nitric oxide. These results are shown in new **Fig. 4e** (please also refer to point 5 of the same reviewer).

3. Fig. 3e requires the addition of complete statistical information, and Fig. 6d should include the corresponding insulin curve, which can be placed in the supplementary materials.

We incorporated the full set of statistical information for **Fig. 3e**. Insulin levels corresponding to **Fig. 6d** are presented in new **Fig. 6a**.

4. The sentence in line 255 “the control mice with parent cell implant” is not clear.

We clarified this sentence.

5. Page 6, the subtitles in line 175 and line 189 are the same. This is misleading.

We corrected this mistake.

6. The figure caption in Fig. 2c “DETA NONOate” should maintain consistency.

We corrected this inconsistency.

7. Please clarify specific advantages of hNORM in line 353.

We highlighted the advantages of hNORM in the revised version.

8. The authors should provide the related amino acid or DNA sequences of hNORM.

In preparation for publication, we will submit the DNA sequences corresponding to the relevant hNORM components to GenBank and will include the accession numbers in the methods section of the final version, ensuring transparency and accessibility of our genetic data for the scientific community.

Reviewer 2

The paper by Mahameed et al. presents the design and optimization of a human nitric oxide-responsive transgene regulation modality (hNORM) in mammalian cells. The researchers successfully established a stable hNORM-transgenic human cell line (HEKhNORM1) and later improved its performance by engineering another cell line (HEKhNORM2) for additional ALDH2 expression. Utilizing topical nitroglycerin patches to control NO production, authors demonstrated precise control of transgene expression in subcutaneously implanted microencapsulated cells in mice. In a proof-of-concept experiment, they showed that hNORM-controlled GLP-1 expression restores blood-glucose homeostasis in a mouse model of type-2 diabetes without causing cardiovascular interference.

While the conceptual framework is intriguing, several critical aspects necessitate further elucidation.

Degree of Advance: The work presents a technological and methodological advancement in gene regulation for therapeutic purposes. However, the translational readiness of this system, particularly concerning long-term safety, drug interactions, and scalability, appears lacking when compared to the existing state of the art in the field.

1. The manuscript lacks quantitative PCR (qPCR) or RNA sequencing data to confirm the expression levels of sGC, PKG1, NLuc, and ALDH2.

In the revised version of the manuscript, we provide qPCR-based expression levels of sGC, PKG1, NLuc, and ALDH2 in hNORM-transgenic cells as requested and these new data are presented in **Fig. S3**. We also generated RNAseq data for both implanted cells and tissues surrounding the implant site (please refer to our response to point 2 raised by reviewer 1). In this way, we provided a thorough understanding of the molecular dynamics and expression profiles associated with hNORM functionality.

2. Although cell viability is mentioned, explicit cytotoxicity assays under various conditions are missing.

We have quantified cell viability in response to DETA NONOate as depicted in **Fig. 3e**. To further validate our findings, we replicated this experiment using a gold standard cytotoxicity assay (CyQUANT™ Cytotoxicity Assay Kit, ThermoFisher Scientific, V23111). This assay relies on the fluorescence-mediated quantification of cytosolic glucose-6-phosphate dehydrogenase (G6PD) released from cells when the plasma membrane is compromised. This new data is presented in **Fig. 3f**.

3. A comparative evaluation of SEAP and NLuc to validate the switch in sensitivity and specificity is lacking. This analysis could provide a stronger rationale for the choice of NLuc.

We conducted this experiment (as depicted in **Fig. 2b**), using both SEAP and NLuc reporter plasmids in different amounts to further explore the detection sensitivity and specificity. This enabled a comprehensive side-by-side comparison of hNORM performance in response to various NO donors, such as DETA NONOate. This new data set is presented in **Fig. S1**.

4. The manuscript displays good reversibility to NO donors but does not elucidate the precise response time for system activation and deactivation. Detailed kinetics concerning the hNORM system's response to NO are important for assessing its clinical utility.

We agree on this important point. Since hNORM is regulated by NO at the transcription level, a precise evaluation of its kinetics would better rely on the quantification of mRNA of the target gene following NO addition or removal. In the revised version of the manuscript, we performed time-lapse qPCR to monitor the activation and deactivation kinetics of this system using hNORM2 cells following DETA NONOate treatment and washout. This experiment is presented in **Fig. S2**. In addition, performed time-lapse live microscopy using fluorescence protein to support our results (see Supplementary Video 1-4 and SI Guide). Furthermore, in-vivo expression kinetic experiment is shown in **Fig. 4g**.

5. While the authors claim the cell line is stable, the manuscript lacks data on long-term stability across multiple passages.

To confirm the stability of hNORM-transgenic HEK-293 cells, we assessed the expression levels of hNORM components, including sGC, PKG1, ALDH2, and GLP-1 by quantitative PCR (qPCR) throughout 30 consecutive passages, covering approximately 3 months of continuous cultivation. These new data are presented in **Fig. S13**.

6. Given ALDH2's role in drug metabolism, how would the concurrent administration of drugs that inhibit or induce ALDH2 affect your system?

This is an intriguing question that we had not previously considered. To investigate this issue, we administered disulfiram, a clinically licensed ALDH2 inhibitor utilized in the treatment of chronic alcoholism (see Koppaka et al., 2012, *Pharmacol Rev.* 64:520), and Alda-1, a compound enhancing the enzymatic activity of ALDH2 that is currently in preclinical development for the treatment of the alcohol flush reaction (Chen et al., 2008, *Science* 321(5895):1493). In addition, we also investigated the effect of commonly used drugs indicated for metabolic syndrome (metformin, cilazapril, atorvastatin, atenolol, aspirin). This experiment was conducted in mice implanted with the hNORM-regulated GLP-1 system treated with NG patch along with these compounds given intraperitoneally. This new in vivo experiment is presented in **Fig. S9** (please also refer to point 12 of the same reviewer).

7. The incorporation of additional enzymes augments the system's complexity which could impede its use and optimization.

The incorporation of ALDH2 is not obligatory, since basic hNORM demonstrates ample sensitivity to clinically licensed doses of nitric oxide, yielding therapeutically effective GLP-1 levels for the treatment of experimental type-2 diabetes and obesity (see **Fig. 4b**). Nevertheless, the enhancement of nitric oxide sensitivity through ALDH2 engineering remains an intriguing preemptive strategy, which could address potential requirements in the future.

hNORM stands out as a streamlined control modality, encompassing only three essential components: a nitric acid sensor, PKG1, and a nitric oxide-driven transgene expression unit. This is in line with several preclinical gene switches that employ three or more components, for example, Adachi et al., 2018 *Nature Biotechnology* 36: 346 (3 components); Cho et al., 2021 *Nature Communications* 12:1 (3 and 4 components); Bouquet et al., 2023 *Gene Therapy* 30: 706 (3 components).

8. The paper lacks discussion on how this system could be scaled for industrial or clinical applications.

We added a comment on this point in the discussion part.

9. The in vivo study covers 35 days, but long-term effects of the system remain unknown. Do the cells maintain their functionality over extended periods?

We have assessed the expression stability and functionality of hNORM in transgenic human cells throughout an extended cultivation period spanning over 30 passages and exceeding 3 continuous months (please refer to our response to point 5 above). In the revised manuscript, we also conducted (i) detailed qPCR profiling of all essential hNORM components (**Fig. S3**) (addressing points 3 and 5 raised by reviewer 1), (ii) comprehensive RNAseq analysis (**Fig. S12**) (addressing point 2 from reviewer 1), (iii) viability assessment (**Fig. S11**) (considering point 4 by reviewer 1, minor point 1 by reviewer 2 and point 2 by reviewer 3), (iv) ALDH enhancer/inhibitor evaluation (**Fig. S9**) (point 6 from reviewer 2), and (v) immuno-histological analyses (**Fig. S11**) (point 4 from reviewer 1), and (vi) fibrosis assessment (**Fig. S11**) (point 4 from reviewer 1 and minor point 1 from reviewer 2) of implanted hNORM-transgenic cells and surrounding tissues at the onset and conclusion of the 35-day experimental period.

The 35-day duration is widely regarded as long-term for the purpose of in vivo analysis. We have compellingly demonstrated that hNORM effectively corrects type-2 diabetes by restoring and maintaining blood-glucose dynamics and homeostasis, as well as addressing obesity by resetting and maintaining normal body weight. This comprehensive characterization of hNORM during the treatment period aligns with other noteworthy proof-of-concept studies targeting prevalent experimental human diseases, many of which have employed similar or even shorter treatment durations (for example, Bai et al., 2019, *Nature Medicine* 25:1266, **35 days**; Huang et al., 2023, *Nature Metabolism* 5:1395, **35 days**; Kemmer et al., 2010, *Nat. Biotechnol.* 28:355, **7 days**; Krawczyk et al., 2020, *Science* 368:993, **7 days**; Liu et al., 2018, *Cell* 174:259, **3 days**; Stanley et al., 2012, *Science* 336:604, **<1 day**; Stanley et al., 2016, *Nature* 531:647, **<1 day**; Wang et al., 2018, *Nature Biomedical Engineering* 2:114, **21 days**; Weber et al., 2002, *Nat. Biotechnol.* 20:901; **3 days**; Weber et al., 2004, *Nat. Biotechnol.* 22:1440, **3 days**; Xie et al., 2016, *Science* 354:1296, **21 days**; Ye et al., 2017, *Nature Biomedical Engineering* 1:5, **20 days**; Ye et al., 2011, *Science* 332:1565, **2 days**; Zhao et al., 2023, *Lancet Diabetes & Endocrinol.* 11:637, **7 days**)

Considering the current ethical and legislative environment in the European Union and Switzerland (highlighted in our response to point 3 of reviewer 1), we feel a complete replication and extension of the 35-day study is not feasible. However, we have comprehensively refined the current in vivo study, as outlined above.

10. The study does not specify the stability or longevity of the encapsulation materials (alginate-PLL-alginate capsules). Can these capsules maintain their structural integrity and functionality over a long period?

We opted for alginate-poly-(L-lysine)-alginate capsules to house human-derived hNORM designer cells within mice based on the widespread use of alginate in previous studies of its efficacy in safeguarding implanted cells, as well as in proof-of-concept studies (for example, Joki et al., 2001, *Nat. Biotechnol.* 19:35; Ye et al., 2011, *Science* 332:1565; Kemmer et al., 2010, *Nat. Biotechnol.* 28:355; Bacchus et al., 2012, *Nat. Biotechnol.* 30:991; Weber et al., 2004, *Nat. Biotechnol.* 22:1440; Rössger et al., 2013, *Nat. Commun.* 4:2825; Schukur et al., 2015, *Sci. Transl. Med.* 7:318ra201; Saxena et al., 2016, *PNAS* 113:1244; Folcher et al., 2014, *Nat. Commun.* 5:5392; Bai et al., 2016, *J. Hepatol.* 65:84; Shao et al., 2017, *Sci. Transl. Med.* 9:eaal2298; Bai et al., 2019, *Nature Medicine* 25:1266; Liu et al., 2018, *Cell* 174:259; Krawczyk et al., 2020, *Science* 368:993; Wang et al., 2018, *Nature Biomedical Engineering* 2:114; Xie et al., 2016, *Science* 354:1296; Ye et al., 2017, *Nature Biomedical Engineering* 1:5; Huang et al., 2023, *Nature Metabolism* 5:1395; Zhao et al., 2023, *Lancet Diabetes & Endocrinol.* 11:637), and in human clinical trials (Jacobs-Tulleneers-Thevissen et al., 2013, *Diabetologia* 56:1605; ClinicalTrials.gov, e.g., NCT01379729, NCT00790257, NCT01739829).

In this proof-of-concept study, the longevity of the implanted hNORM cells was not specifically examined. However, it has been firmly established that (engineered) cells, when implanted in immuno-protective alginate beads, exhibit a propensity to proliferate until they reach a state of contact inhibition. Consequently, these cells can remain functional over extended periods of time. Noteworthy reports validating this phenomenon include (i) in rodents: Lamb et al., 2011, *Transplant Proc.* 43:3265 (8 weeks); An et al., 2018, *PNAS* 115:E263 (4 months); Veiseh et al., 2015, *Nat. Materials* 14:643 (180 days); Vegas et al., 2016, *Nat. Med.* 22:306 (174 days); de Vos et al., 2003, *Biomaterials* 24:305 (200 days); Schneider et al., 2005, *Diabetes* 54:687 (>7 months); Qi et al., 2012, *Xenotransplantation* 19:355 (232 days); Duvivier et al., 2001, *Diabetes* 50:1698 (>350 days); (ii) in primates: Vegas et al., 2016, *Nat. Biotechnol.* 34:345 (>6 months); (iii) in humans: Jacobs et al., 2013, *Diabetologia* 56:1605 (3 months); Soon et al., 1994, *Lancet* 343:950 (9 months); Groth et al., 1994, *Lancet* 344:1402 (200-400 days); Calafiore et al., 2006, *Diabetes Care* 29:137 (1 year); Elliott et al., 2000, *Cell Transplant.* 9:895 (2 years); Tuch et al., 2009, *Diabetes Care* 32:1887 (2.5 years); Basta et al., 2011, *Diabetes Care* 34:2406 (3 years); Soon et al., 1999, *Adv. Drug Deliv. Rev.* 35:259 (58 months); Elliott et al., 2007, *Xenotransplantation* 14:157 (9.5 years).

In the revised manuscript, we analyzed the capsule integrity and functionality as presented in **Fig. S11** and **Fig. S12** after 35 days as described in **Fig. 5b**. Crucially, as illustrated in **Fig. 5b**, we found that alginate-encapsulated hNORM-transgenic HEK-293 cells exhibited responsiveness to nitric oxide and consistently released therapeutic levels of GLP-1 throughout the entire 35-day study period, supporting the stability of these implants.

11. There is no mention of whether these capsules elicit a foreign body reaction, which is a key concern in any implantable device. Any reaction could affect the function of the encapsulated cells and therefore the efficacy of the system.

As noted in our response to point 10, alginate has been widely used in experimental and clinical applications. Further, a meticulous examination of long-term viability and fibrosis associated with alginate-encapsulated HEK-293 cells in mice was conducted in a recent 5-week study (Bai et al., 2019; *Nature Medicine* 25:1266 and Krawczyk et al., 2020, *Science* 368:993). To confirm the lack of immunogenicity of the present capsules, we conducted an extensive systemic cytokine profiling (**Fig. S10**), immuno-histological analyses (**Fig. S11**), fibrosis assessment (**Fig. S11**), and RNAseq (**Fig. S12**) of both the implanted cells and the surrounding tissue.

12. Since many type-2 diabetes patients are on multiple medications, it would be useful to study how your system interacts with these commonly used drugs. This is particularly important given the central role of ALDH2 in drug metabolism.

To investigate this issue, we administered disulfiram, a clinically licensed ALDH2 inhibitor utilized in the treatment of chronic alcoholism (see Koppaka et al., 2012, *Pharmacol Rev.* 64(3):520), and Alda-1, a compound enhancing the enzymatic activity of ALDH2 that is currently in preclinical development for the treatment of the alcohol flush reaction (Chen et al., 2008, *Science* 321(5895):1493). In addition, we also investigated the effect of commonly used drugs indicated for metabolic syndrome (metformin, cilazapril, atorvastatin, atenolol, aspirin). This experiment was conducted in mice implanted with the hNORM-regulated GLP-1 system treated with NG patch along with these compounds given intraperitoneally. This new in vivo experiment is presented in **Fig. S9** (please also refer to point 6 of the same reviewer).

Minor Comments:

1. No histological analyses of the explanted encapsulated cells are provided. This could show how

the system interacts with surrounding tissues and whether it induces any adverse reactions or fibrosis.

In the revised version, we performed a comprehensive histological analysis (**Fig. S11**), encompassing systemic cytokine profiling **Fig. S10**, along with immuno-histological examinations and a fibrosis assessment of both the implanted cells and the surrounding tissue. Please refer to our responses to reviewer 1's point 4 and this reviewer's point 11. The methodology is based on recent work by Bai et al. (2019; *Nature Medicine* 25:1266) and Krawczyk et al. (Krawczyk et al., 2020, *Science* 368:993), in which a meticulous examination of the long-term fibrosis associated with alginate-encapsulated HEK-293 cells in mice was carried out.

2. While the discussion talks about the advantages and the broader context, it doesn't adequately discuss the limitations of the study itself. For example, the dependency on ALDH2 and its implications are not discussed here.

We have addressed the limitations inherent in this study as the reviewer suggested.

3. A more detailed comparison to existing therapies, especially in terms of efficacy, safety, and cost-effectiveness, would strengthen the discussion.

We addressed these points in the revised discussion section.

4. While the paper talks about how the system could be used in chronic conditions, it does not discuss how it compares to existing treatments for those conditions in terms of effectiveness, safety, and cost.

We addressed these points in the revised discussion section.

Point-by-point responses to Reviewer 3.

In this study, the authors have developed a new method, referred to as human Nitric Oxide-Responsive Transgene Regulation Modality (hNORM), which allows for the controlled production and distribution of therapeutic proteins in response to NO. This response is triggered by the enzymatic conversion of nitroglycerin (NG), delivered percutaneously through clinically approved patches. To demonstrate its feasibility, they applied this system to deliver GLP-1 through subcutaneously implanted microencapsulated hNORM-transgenic GLP-1-expressing human cells controlled by licensed dermal patches placed above. This device offers intriguing potential for the modulation of targeted protein expression via an NG patch. Nevertheless, several concerns must be addressed before potential acceptance.

1. The system exhibits a protracted response time, constraining its utility. Would the authors consider proposing a more viable application scenario beyond GLP-1 deployment?

As a gene switch modulating transgene expression, hNORM exhibits a response time comparable to other transcription-based transgene control modalities. Noteworthy examples of transcription-based gene switches validated in vivo include Ye et al., 2011, *Science* 332:1565; Kemmer et al., 2010, *Nat. Biotechnol.* 28:355; Weber et al., 2004, *Nat. Biotechnol.* 22:1440; Rössger et al., 2013, *Nat. Commun.* 4:2825; Schukur et al., 2015, *Sci. Transl. Med.* 7:318ra201; Folcher et al., 2014, *Nat. Commun.* 5:5392; Bai et al., 2016, *J. Hepatol.* 65:84; Shao et al., 2017, *Sci. Transl. Med.* 9:eaal2298; Bai et al., 2019, *Nature Medicine* 25:1266; Liu et al., 2018, *Cell* 174:259; Wang et al., 2018, *Nature*

Biomedical Engineering 2:114; Xie et al., 2016, Science 354:1296; Ye et al., 2017, Nature Biomedical Engineering 1:5, and Huang et al., 2023, Nature Metabolism 5:1395.

Validation of hNORM-based transcription control has been conducted using the reporter gene nanoluciferase (NLuc), the human model glycoprotein SEAP (human placental secreted alkaline phosphatase), and the therapeutic peptide hormone GLP-1 (glucagon-like peptide 1). We posit that hNORM control has the potential to finely regulate the expression of any desired target gene.

Several high-profile proof-of-concept studies have selected GLP-1 expression as the gold standard for therapeutically validating gene switches, as demonstrated by Ye et al., 2011, Science 332:1565; Ye et al., 2013, PNAS 110:141; Xie et al., 2016, Science 354:1296; Xue et al., 2017, Mol. Ther. 25:443; Shao et al., 2017, Science Transl. Med. 9:eaal2298; Bojar et al., 2018, Nature Communications 9:2318; Mansouri et al., 2021, Nature Communications 12:3388; Guha Ray et al., 2023, Nature Communications 14:3151.

To facilitate a comparative analysis of percutaneous transgene modalities, we directly compared the response dynamics of hNORM with those of transdermal phloretin (Gitzinger et al., 2009, PNAS 106:10638) and menthol (Bai et al., 2019, Nature Medicine 25:1266) transgene control modalities. As we anticipated, the rapid diffusion of nitric oxide through membranes may confer a competitive advantage in terms of speed. These new data are presented in **Fig. S8** which show that hNORM stands well along with other transcription-based gene switches.

We wish to emphasize that our selection of GLP-1 for validation of our novel nitroglycerin/nitric-oxide-tunable gene switch is strategic. GLP-1 and its derivatives, including the blockbuster drugs Ozempic, Saxenda, and Wegovy, are pivotal in managing type-2 diabetes and obesity. These conditions have a global impact, affecting over 10% of the population today, as reported by the International Diabetes Federation (diabetesatlas.org). Moreover, their prevalence is expected to rise, with nearly one-quarter of the human population projected to be affected by 2035, according to the World Obesity Atlas 2023 (www.worldobesity.org).

We incorporate these points in the discussion section of our revised manuscript. (please refer to point 1 of reviewer 1)

2. The survival metrics of HEK-hNORM cells post-implantation remain unquantified. It is imperative to determine whether these cells succumb to apoptosis or other cell death over extended periods post-implantation and to assess the subsequent impact on the immune microenvironment at the implantation site and the broader immune system response.

We have substantiated that hNORM-transgenic HEK-293 cells effectively rectify hyperglycemia associated with type-2 diabetes and restore a normal body weight throughout the entire study period (see **Fig. 5b** and **Fig. 6 a-d**). Consequently, it is reasonable to infer that the implanted cells maintain their functionality during this period. Please also refer to our response to point 10 of reviewer 2 – there is extensive evidence that alginate-encapsulated engineered cells remain viable for long periods.

To confirm this, we conducted a comprehensive viability/apoptosis profiling and RNAseq of the implanted hNORM-transgenic HEK-293 cells at both the commencement and conclusion of the 35-day study period, addressing the concerns raised in point 4 by reviewer 1 and minor point 1 from reviewer 2. These new data are presented in **Fig. S11** and **Fig. S12**, respectively. Specially, we conducted further assessments of cell viability and implant fibrosis (according to Krawczyk et al., 2020, Science 368:993; Bai et al., 2019, Nature Medicine 25:1266), as well as immuno-histochemical analyses (according to Huang et al., 2023 Nature Metabolism 5: 1395; Krawczyk et al., 2020, Science 368:993; Maity et al., 2023, Advanced Materials 35:e2300890) at both the initiation and conclusion of the 35-day experimental study. These histological data are presented in supplemental **Fig. S11** and these results show (i) insignificant local immune response, (ii) absence of fibrosis, (iii) negligible apoptosis of the encapsulated cells (as shown by anti-caspase-3 immunohistochemistry), and (iv) high integrity and stability of the capsules.

3. The selection of HEK-293 cells for in vivo implantation raises concerns. The authors are urged to conduct comprehensive analyses to confirm the absence of systemic toxicity and to rule out local immune cell infiltration at the implantation site in the treated animals. Furthermore, it is crucial to investigate whether there are discernible differences in the NO response in cells pre- and post-implantation.

Although unlikely to be adopted for clinical applications, HEK-293 cells have emerged as the gold standard human cell line for validating proof-of-concept studies in the treatment of experimental human diseases. Numerous high-impact publications highlight novel, prototypical cell-based therapies employing HEK-293. Notable examples include Bai et al., 2019, *Nature Medicine* 25:1266; Huang et al., 2023 *Nature Metabolism* 5: 1395; Kemmer et al., 2010, *Nature Biotechnology* 28:355; Liu et al., 2018, *Cell* 174:259; Krawczyk et al., 2020, *Science* 368:993; Stanley et al., 2012, *Science* 336:604; Stanley et al., 2015, *Nature Medicine* 21:92; Stanley et al., 2016, *Nature* 531:647; Wang et al., 2018, *Nature Biomedical Engineering* 2:114; Xie et al., 2016, *Science* 354:1296; Ye et al., 2017, *Nature Biomedical Engineering* 1:5; Ye et al., 2011, *Science* 332:1565; Zhao et al., 2023, *Lancet Diabetes and Endocrinology* 11:637; Zhou et al., 2022, *Nature Biotechnology* 40:262.

Both our research and others' work have extensively documented on multiple occasions that encapsulated engineered cell lines, including HEK-293, do not induce either local (Krawczyk et al., 2020, *Science* 368:993; Rössger et al., 2013, *Nat. Commun.* 4:2825) or systemic inflammation, or immune cell infiltration (Bai et al., 2019, *Nature Medicine* 25:1266; Huang et al., 2023 *Nature Metabolism* 5: 1395). To substantiate these findings in the context of hNORM, we conducted a comprehensive profiling of cytokines in addition to immuno-histochemical analyses of the implant and its surrounding tissue at the beginning and the conclusion of the 35-day study period, as suggested in point 4 by Reviewer 1 and point 11 by Reviewer 2. These new data are presented in **Fig. S10** and **Fig. S11**.

Minor issues:

4. The manuscript exhibits several formatting inconsistencies. Specifically, the indentation of paragraphs, such as those beginning on lines 154 and 177, is irregular; line 189 contains a superfluous subheading; the abbreviation "Fig" in line 258 is not appropriately bolded; and the citation format for articles, as seen in references 6, 25, 31, and 43, deviates from the standard.

Thank you for your comment. We fixed these issues in the revised manuscript.

5. The captions of figures do not conform to accepted standards and necessitate revision, such as "Figure 3 | hNORM is effective, tunable, and reversible, and is blocked by methylene blue."

We corrected this in the revised manuscript.

Reviewer 4

The authors describe a nitric oxide (NO)-responsive transgene regulation modality (hNORM) to control the dynamic production and delivery of protein therapeutics. The strategy is based on engineered enzymatic reaction platform converting nitroglycerin (NG), that is transdermally delivered by clinically licensed patches, to NO. The authors have previously reported a series of in situ transgene expression & switching systems, based upon synthetic biological engineering approach, that are potentially relevant to the future gene- and cell-based therapies. In the present report, the essence lies on the choice of NG, which has been widely used to control heart conditions, in combination with a clinically licensed NG patches. To me, this is a wise and convincing combination

in the context of clinical translation. Overall, the manuscript is written well including the concept, design, experimental detail and results and properly discussed. I did not find any particular technical issues in the manuscript. Therefore, I recommend the paper be considered publication in Nature Biomedical Engineering after some improvements.

I have two suggestions to strengthen the paper:

1. There is no real image shown at all, which makes it a bit less eye-appealing. Please consider to provide some for device specification, implantation site, in vivo or ex vivo NG biodistribution imaging, and etc.

We are grateful for this comment. In the revised manuscript, we provided a fluorescence microscopy-based time-lapse movie to illustrate the real-time kinetics of the hNORM system (see Supplementary Video 1-4 and SI Guide) in addition to comprehensive immuno-histochemical analyses at the start and the endpoint of the 35-day study period (**Fig. S11**) (please see point 4 of reviewer 1 and point 11 of reviewer 2).

2. In the discussion please provide any limitations & issues to be addressed before clinical translation.

We included these points in the revised manuscript.

Minor issue:

1. 143: Should “PGK2-expressing” be “PKG2-expressing”?

Thank you for noting this mistake. We fixed it in the revised manuscript.